# Person Recognition via Gait: A Review of Covariate Impact and Challenges

**DOI:** 10.3390/s25113471

**Published:** 2025-05-30

**Authors:** Abdul Basit Mughal, Rafi Ullah Khan, Amine Bermak, Atiq ur Rehman

**Affiliations:** 1Department of Computer Science, Bahria University, Islamabad P.O. Box 44000, Pakistan; sardarbasitmughal344@gmail.com; 2Department of Electrical & Computer Engineering, Pak Austria Fachhochschule: Institute of Applied Sciences and Technology, Haripur P.O. Box 22620, Pakistan; rafi.ullah@paf-iast.edu.pk; 3Division of Information and Computing Technology, College of Science and Engineering, Hamad Bin Khalifa University, Doha P.O Box 5825, Qatar

**Keywords:** gait, covariates, deep learning, computer vison, gait identification

## Abstract

Human gait identification is a biometric technique that permits recognizing an individual from a long distance focusing on numerous features such as movement, time, and clothing. This approach in particular is highly useful in video surveillance scenarios, where biometric systems allow people to be easily recognized without intruding on their privacy. In the domain of computer vision, one of the essential and most difficult tasks is tracking a person across multiple camera views, specifically, recognizing the similar person in diverse scenes. However, the accuracy of the gait identification system is significantly affected by covariate factors, such as different view angles, clothing, walking speeds, occlusion, and low-lighting conditions. Previous studies have often overlooked the influence of these factors, leaving a gap in the comprehensive understanding of gait recognition systems. This paper provides a comprehensive review of the most effective gait recognition methods, assessing their performance across various image source databases while highlighting the limitations of existing datasets. Additionally, it explores the influence of key covariate factors, such as viewing angle, clothing, and environmental conditions, on model performance. The paper also compares traditional gait recognition methods with advanced deep learning techniques, offering theoretical insights into the impact of covariates and addressing real-world application challenges. The contrasts and discussions presented provide valuable insights for developing a robust and improved gait-based identification framework for future advancements.

## 1. Introduction

Humans have always looked for a number of different measures to protect their systems; they use password protection, encryption keys, and other means of protection from unauthorized access. With the advancement in technology, human identity is increasingly being utilized across a growing number of applications for authorized access. Biometrics features mean the use of essential physical or psychological traits for the identification of humans [1]. Biometrics are typically categorized into two types [2]: behavioral and physical. Behavioral biometrics cover signature analysis, keystroke dynamics, voice recognition, and gait recognition, and physical biometrics comprise facial recognition, retinal identification, and iris scanning. Human gait is one of the most repetitive collaborations between numerous human body parts and patterns. Gait-based human recognition is a biometric technique that permits identifying a person from a distance. Gait-based human recognition is mainly focused on the walking style of the individual. Gait recognition is another example of a non-intrusive biometric that recognizes a person from a distance [3]. It provides a considerable benefit compared to other approaches because it can observe and identify a person in low-quality videos, which few other biometric systems can do [4]. Moreover, it also differentiates itself from other biometrics because of its capability of remote collection through surveillance camera, floor sensors, and so on without the subject being aware of it.

Gait-based recognition systems allow users to create a system through which users automatically register a person’s movement and gait algorithms, which can serve to analyze movements. Gait-based authentication approaches will create a more comfortable authentication method for users. Gait identification is one of the most advanced and future-generation recognition technologies. The structure of this manuscript is outlined in Figure 1.

The steps in a gait identification system [5] are as follows: Data collection is the initial phase in a gait recognition system; to recognize a person one needs to have data about gait parameters. As for that purpose, several techniques can be applied, among which are video recording, pressure sensors, floor sensors, and motion capture cameras. Preprocessing is the second step in the gait identification system, where noise is removed from the input images, and background subtraction is performed in appearance-based models. Feature extraction is the third step in identifying an individual. This step involves collecting unique characteristics, such as walking patterns, walking speed, and foot angle. Dimension reduction is the fourth step, in which the dimensions of features are reduced. This step is important because the features that have been derived using the unprocessed data cannot be utilized in classification. The unprocessed data are usually rich with an abundance of feature variables, and their quantity may be several times greater than the training sample size. The final phase in human gait recognition systems is classification, where a person is recognized from their gait features obtained in the previous steps. The method of feature extraction is accomplished through ML and DL methods. A flow diagram of the gait recognition system is described in Figure 2.

Human gait analysis systems are primarily categorized into two methods: model-based and model-free [6]. Model-based methods include mathematical models used in describing a person’s walking motion, with importance on the joints’ angles of motion through walking. These techniques were intended to study the basic dynamics of gait in terms of joint motion. Model-free strategies alternatively use extracted features from the observed walking behaviors, such as the body structure and limb movement. This involves the tracking and quantification of the human silhouette obtained from gait sequences to establish different walking patterns [7]. As model-free approaches do not need additional sensors or subjects’ permission, they operate solely on the video frames extracted from the cameras. These properties make them very useful in real-world applications. Despite the strengths of model-based approaches, where such strategies have benefits, there are also several drawbacks with the use of their foundations, including an excessive load on resources and the incorrect estimation of keypoint approximation [8,9,10]. For this reason, model-based methods perform less appropriately in the recognition tasks when compared to the appearance-based method. These limitations have indeed led to extensive research and have contributed to the acceptance of appearance-based methods as the most common methods in the field.

Covariates are the factors in the gait identification system that affect the accuracy of the system. These covariate factors include clothing variations, view angles, walking speeds, occlusion, and lighting conditions. With the advancement in technology, model-based technologies, along with model-free and fusion of model-based and model-free approaches, are utilized to handle covariate conditions. Advanced computer vision techniques like the Lower–Upper Generative Adversarial Network are used to create 3D pose sequences from input frames. LUGAN consists of Graph Convolutional Networks and Fully Connected Networks to extract better gait features. The model is trained using the CASIA and OUMVLP datasets to handle view angles ranging from 0 degrees to 270 degrees, along with clothing variation, hand carrying, and normal walking [11]. Advanced object detection methods like YOLOv2 and SqueezeNet are utilized to extract deep gait features with the help of the FireConcat-02 layer. The Bi-LSTM model is used with YOLO to extract spatiotemporal features that efficiently and effectively represent the gait features. The LSTM model with 1000 hidden units and a dropout rate of 0.5 enabled the gait model to perform better under a variety of covariates, including clothing variation, view angles, and walking speeds [12]. Virtual dressing is a technique in which digital clothes are applied to a 3D model to analyze how different clothes affect the accuracy of the gait model. This helps train the model to perform better under unknown covariate conditions. Human 3D poses are extracted by utilizing a body parsing method, which segments the human body into different parts to obtain meaningful information from specific regions. With the help of the advanced Hierarchical Temporal Memory network, the model learns the best gait patterns from the segmented regions. To enhance the model’s performance under diverse covariate conditions, a top–down attention mechanism method is applied [5]. Fusion-based techniques for gait recognition are becoming increasingly popular because these methods handle covariate factors more effectively and efficiently, as they utilize the properties of skeleton-based features and Gait Energy Image (GEI) features. The Skeleton Gait Energy Images combined with the GEI enhances the extraction of gait features and effectively address challenges such as view angle changes and clothing variations. The cascade CNN model extracts skeleton joint information in the initial stage of the CNN, generating confidence maps to identify body parts, while the other network generates part affinity fields that represent the relationships between different body parts. Skeleton Gait Energy Images are created from skeleton keypoints, and these keypoints are expanded into specific widths and sizes to form a human model. The Hybrid Descriptor combines both GEI and SGEI to improve gait recognition. To address covariate factors, a special block called the View Transformation Model is introduced. This model decomposes gait characteristics into subject-independent and view-independent matrices. Similarity is calculated using Euclidean distance, combining the distances for both GEI and SGEI [13]. Another fusion-based technique in which human joint information is obtained from pose sequences uses the Mediapipe algorithm. The pose sequences contain static features, such as the distance between different body joints, like the distance between the left and right knees. These static features provide spatial information, while dynamic features, such as the knee angle during walking, provide angular information of the body joints. The features from appearance-based and skeleton-based models are fused into a single feature vector, which contains rich information about the gait sequence. For classification, the concatenated feature vector is passed through fully connected layers to recognize an individual. The model consists of various fully connected layers, with the final layer being the output layer using a SoftMax classifier. The model is trained on the CASIA dataset. The model effectively handles different covariate conditions, such as clothing variations, view angles, and walking speeds. The model demonstrates excellent accuracy in recognizing individuals under clothing variations and varying view angles, but the computational time of the model increases as a result [14].

The significant contributions of the manuscript are as follows:This research discovers and investigates numerous gait recognition techniques under different covariates conditions, including both traditional approaches and advanced machine learning approaches, while underlining their respective limitations.It describes data repositories used for gait recognition, along with their limitations and challenges.It outlines common evaluation metrics usually employed as standard practice and specifies which metrics are most effective for addressing different covariates.It also provides an in-depth analysis of gait identification methods, examining their performance across various datasets, along with their advantages and disadvantages.It also outlines the theoretical modeling of covariate impacts on the gait identification system, addressing real-world applications and deployment challenges in existing surveillance methods and actionable future research recommendations.

This article is outlined as follows: Section 2 offers a detailed summary of the source databases and discusses the standard evaluation metrics widely utilized in the field. Section 2 also describes and discusses various techniques used in gait recognition systems. Section 3 offers a comprehensive comparison of gait recognition performance under various covariate factors that have been explored. The section also contains an analysis of the studies, along with their limitations and constraints. Section 4 offers future directions and provides a conclusion for the gait recognition system.

## 2. Materials and Methods

Gait identification is one of the most significant examples of biometric identification tasks. In gait identification, extraction of the human pose from a distance, caught by a camera, and identification of the person make it unique from other biometric techniques. In the field of computer vision, human identification across multiple cameras is a significant challenge; especially in person monitoring systems, the identification of individuals from various groups of pictures captured by numerous cameras is known as re-identification of the person. Covariate factors can reduce the accuracy of the gait recognition system. Gait recognition is an emerging research field, as many datasets include various covariate factors and different view angles. Many existing studies use deep learning models to address covariate factors; however, these models lack important capabilities, as no single model can effectively handle all covariates, such as clothing variations, view angles, speed variations, and occlusion. While some research attempts to address these issues, it often results in increased computational time and decreased accuracy. We selected peer-reviewed papers and conference publications that specifically consider these covariate factors. The selection of records for the manuscript was based on the following criteria outlined below. The Prisma2020 guidelines [15] were followed to conduct the review. A detailed PRISMA diagram, Figure A1, is attached to the Appendix A of this paper.

### 2.1. Search Databases

The papers that fit this review use (a) RGB, silhouette, thermal, and pose images or videos; (b) include covariate factors such as view angles, clothing variations, walking speeds, carrying conditions, and low-light conditions; and (c) incorporate deep learning techniques, either model-based, model-free, or a fusion of both. We used keyword searching techniques across numerous online databases, including MDPI, IEEE Xplore Digital Library, Elsevier Library, PLOS, and Springer Nature.

### 2.2. Filtering Criteria

We considered only academic conference and peer-reviewed journal papers written in English. The study focuses solely on gait biometrics and emphasizes covariate factors that reduce the accuracy of gait identification systems. We included papers where recognition is performed using images or videos, excluding those that use sensor data for gait recognition. The remaining papers were filtered using a snowballing strategy. A total of 2310 records were initially identified. Through personal reading, we selected papers published between 2017 and 2024 that addressed at least one covariate factor. From these, 54 of the best papers were chosen for this review study. The PRISMA diagram for the study is shown in Figure 3.

### 2.3. Eligibility Criteria

To select the papers, three methods were employed: reading the title of the research or conference paper, reading the abstract, and reading the full paper. Following this filtering process, 570 articles were shortlisted for further evaluation. In the initial stage, 302 papers were removed after reviewing their titles. From the remaining papers, 101 were excluded after reading their abstracts, as they did not meet the criteria for covariate factors. Finally, after thoroughly reviewing the full text, 54 of the best papers were selected. These papers utilized image and video datasets and included covariate factors, making them most suitable for the objectives of our proposed work.

### 2.4. Inclusion Criteria

In the inclusion method, we selected 54 papers that discuss various factors. The 54 selected papers focus on gait recognition through images and videos and include covariate factors that impact the accuracy of the gait recognition system. We excluded sensor-based gait recognition studies because they rely on controlled environments and do not account for covariate factors. This is because different sensors collect human movement data without considering what the person is wearing, the lighting conditions, or the carrying conditions. We prioritized covariate factors such as clothing conditions, carrying conditions, view angles, low lighting conditions, and different walking speeds, as these are the most common factors that affect the performance of gait recognition systems in real-time environments, such as security surveillance and access control. Most of the records are peer-reviewed papers and conference proceedings published between 2017 and 2024, selected due to the major advancements in computer vision and deep learning during this period. Earlier studies did not include covariate factors and relied on traditional handcrafted techniques.

### 2.5. Data Analysis

This section focuses on scrutinizing and categorizing data for the selected records. The selected papers are analyzed by using two different techniques, and the papers that meet the requirements are considered for review and subsequently evaluated. The number of papers selected per database is shown in Figure 4.

#### 2.5.1. Year-Wise Evaluation

The selected papers were analyzed by year to evaluate the level of scholarly interest in gait recognition over the past few decades. Research on gait identification systems began in the early 2000s, with most advancements occurring after 2015. Among the selected articles, 3 were published in 2017, 5 in 2018, 1 in 2019, 4 in 2020, 8 in 2021, 14 in 2022, 10 in 2023, and 9 in 2024 (see Figure 5).

#### 2.5.2. Type of Selected Articles

Furthermore, the selected articles were subjected to a comprehensive evaluation based on their type. The different types of articles included in this study are depicted in Figure 6. Among the total records, 44 are journal papers, 9 are conference papers, and 1 is an arXiv paper.

In early 350 BC, Aristotle studied animal gait by observing and investigating parameters such as weight, health, size, and age. Likewise, in the study of gait analysis in humans, studies have strived to unveil what gait movement reflects in terms of the physical and behavioral aspects. Within the domain of human gait analysis, Provencher and Abd [16], often regarded as the fathers of biomechanics, who played a pivotal part in advancing the field in the closing years of the seventeenth century, highlighted that the diverse gait patterns are created through the motion of muscles [17]. Weber and Weber in 1836 made a quantitative approach to the temporal and spatial characteristics of gait patterns. Subsequently, Muybridge and Marey in 1874 documented and analyzed human gait and other movement patterns using photographic methods [18]. Gait was recognized as a biometric indicator through video analysis in 1984 [19]. In 2000, the HumanID DARPA dataset was recognized as part of a research program on distance-based identification during the rise of digital video recording technology. Since then, advancements in technology, utilizing 3D cameras, mobiles, accelerometers [20], and sensors, are employed to record gait patterns. More recently, DL methods like CNN [21], Human identification based on RNN, ANN, and cross-view gait models have gained significant popularity within the research community [22]. The history of gait identification is shown in Figure 7.

There are various types of input used in gait recognition systems. The first input type is the RGB image, and the second is the silhouette. A silhouette in gait is a binary map of a person walking, used to identify their gait. The third input type is Gait Energy Images (GEIs), which are considered a robust gait representation for gait-based human identification. In gait recognition, a Gait energy image is usually calculated from one full gait cycle. Additional inputs consist of optical flow, 2D and 3D skeleton data, and 3D mesh techniques, as depicted in Figure 8.

Many methodologies are employed in the literature for gait recognition under different covariate factors, such as CNN models like VGG, ResNet, and AlexNet, as well as some model-based techniques like pose estimation models. Covariates are factors that reduce gait identification system efficiency, such as different wearing conditions, carrying conditions, and view angles from which pictures are taken. These factors affect gait recognition accuracy. Different types of covariate factors are available in datasets, and various models are used to tackle these issues.

### 2.6. Source Database

Numerous datasets for gait recognition have been presented in the literature. These datasets contain different types of covariate factors and various view angles for the training and testing of gait recognition systems. The databases outlined in Table 1 consist of those utilized in these studies. Different databases exhibit a wide range of image variations. Furthermore, a higher level of image variation is useful to build a more general gait recognition model. Also, the total count of subjects present within the dataset determines the set of subjects utilized to train the generalized and durable model. These datasets cover a variety of covariate factors, including view angles, clothing variations, carrying conditions, and different lighting conditions. Numerous datasets contain videos, human silhouettes, RGB images, human poses, infrared images, and 3D human meshes to enhance model performance in different scenarios.

Previously, numerous datasets were utilized to measure the performance of gait recognition systems. However, these databases are constrained in several ways, ranging from differences in appearance, multiple views of individuals, or under different weather conditions, among others. In training deep learning models, structured approaches require organizing datasets that are large in size with a variety of numbers of samples and varying environmental conditions. In data preparation, there are two main issues. First, it is required to record sequences of videos or images of, for instance, an individual performing different movements in the course of the gait cycle. Second, there are questions of ethics regarding the data collected from anybody, either in public places or even in their own private properties, sometimes with their consent.

#### Limitations of Existing Datasets and Recommendations

There are many limitations to the existing dataset, which cause an impact on the performance of the model. The limitations of each dataset are provided below:

***Tum Gait:*** The TUM Gait dataset includes several covariate factors, such as normal walking, where the person walks six times—three times to the right and three times to the left. It also covers walking with a backpack, where the individual carries a 5 kg bag. The third covariate is shoe variation, including an example with coated shoes, where the shoes are covered with plastic. However, the dataset does not include variations in view angles, walking speed, and clothing and carrying conditions, or changes in lighting and weather conditions.

***CASIA Dataset:*** The CASIA dataset is considered one of the largest and most comprehensive databases for gait identification. It covers various covariate conditions such as view angles, walking speeds, and some clothing variations. The CASIA A dataset contains data from 20 people, captured at three different view angles: 0°, 45°, and 90°. The CASIA B dataset is one of the most famous datasets used for gait recognition, as it covers 74 individuals with three covariate factors: normal walking, walking with a bag, and walking with a long coat. These sequences are captured from 11 different view angles, ranging from 0° to 180°. The CASIA C dataset includes infrared images of 153 individuals, captured under nighttime conditions. The dataset also includes individuals carrying a backpack with normal walking sequences and three walking speed variations: Normal Walk (NW), Fast Walk (FW), and Slow Walk (SW). Each subject has 2 Fast Walking, 2 Slow Walking, 4 Normal Walking, and 2 Carrying Backpack sequences. CASIA E, released in late 2020, includes silhouettes of 1014 subjects with variations in walking speeds, such as normal walking, walking while wearing a coat, and walking while carrying a bag. The dataset features frames captured from 15 different view angles.

While the CASIA dataset provides a wide range of covariates, it has certain limitations. The dataset does not contain many clothing variations, such as different pants, shirts, and shoes. The view angles range from 0° to 180°, which is limited. Furthermore, the dataset does not include changes in lighting conditions, different weather conditions, or variations in carrying conditions, which are important for modeling gait recognition in diverse real-world scenarios.

***OUISIR:*** The OU-ISIR dataset is a well-known database in the field of gait recognition, as it includes several versions that cover various covariates known to affect the accuracy of gait recognition systems. The OU-ISIR dataset consists of 4007 individuals, captured at four different view angles: 55°, 65°, 75°, and 85°.

The OU-ISIR Treadmill Dataset includes 32 clothing variations, such as regular pants, baggy pants, short pants, skirts, half shirts, full shirts, and many other variations. The sequences are captured from 25 different view angles, with individuals ranging in age from 15 to 65 years.

The OU-ISIR Speed dataset contains gait silhouettes of 34 subjects walking at different speeds, ranging from 1 km/h to 5 km/h. It provides useful data for evaluating gait recognition under varying walking speeds.

The OULP-Age dataset contains 63,846 GEIs of males and females, with age ranges from 2 to 90 years. The dataset consists of 31,093 male GEIs and 32,753 female GEIs, offering valuable insights into how gait recognition systems perform across different age groups.

The OU-LP Bag dataset includes 62,528 videos of individuals carrying a bag. The dataset contains three sequences: two sequences where individuals are carrying a bag and one sequence where no bag is carried.

While the OU-ISIR dataset is diverse and includes a range of clothing variations and speed conditions, it has limitations. The dataset does not include variations in carrying objects with speed variations. Additionally, the dataset does not contain different lighting conditions, as all its sequences are captured under only one lighting condition. Furthermore, the dataset lacks occluded frames, which reduces the accuracy of the system in real-time situations where partial occlusions are common.

***OU-MVLP:*** The OU-MVLP dataset is considered one of the most famous and largest datasets for gait recognition, as it consists of 259,013 frames. The dataset includes 10,307 subjects, with ages ranging from 2 to 87, captured from 14 different view angles, from 0° to 90° and 180° to 270°. The OU-MVLP-Pose dataset is built from two pretrained models, OpenPose and AlphaPose. This dataset contains pose information of subjects captured from 15 different view angles. The limitations of the dataset are that it does not contain clothing variations, has limited view angles, does not include different walking speeds, and does not account for variations in lighting conditions or occluded frames.

***USF:*** The USF dataset consists of 1870 sequences and includes a limited number of covariates, such as different shoe types, hand carry, and various surface types, including grass and concrete. However, the dataset has a minimal number of viewpoints, clothing variations, walking speeds, and lighting conditions.

***Southampton dataset:*** The Southampton dataset, designed by the University of Southampton, consists of three versions. The initial version of the dataset, known as the SOTON small dataset, consists of 12 subjects with different walking speeds, variations in clothing and shoes, along with hand carry. SOTON large is the second version of the dataset, which consists of 114 subjects with 6 different view angles, walking on a treadmill. SOTON temporal is the third version of the dataset, which includes subjects aged 20 to 55 years, with different backgrounds, lighting conditions, and walking surfaces. The limitation of the dataset is that it does not cover large view angles, clothing variations, walking speeds, or occluded frames.

### 2.7. Evaluation Metrics

Several performance measures are employed in the literature for gait recognition system under different environments. The most common evaluation metrics are Recognition accuracy, Sensitivity, False Positive Rate, Precision, F1-Score, Receiver Operating Characteristic (ROC) Curve, Area Under the ROC Curve (AUC), Equal Error Rate (EER), Cumulative Matching Characteristic (CMC) Curve, and Rank-1 Accuracy.

#### 2.7.1. Sensitivity

Sensitivity is calculated as the ratio of correctly identified subjects or gait instances with respect to the overall samples. Sensitivity is affected by changes in view angles, clothing variations, occlusions, and walking speed; as a result, recall decreases. Sensitivity is only applicable when the aim is to identify as many correct matches as possible, even at the cost of some false positives. It is important in surveillance and forensic analysis.

The formula is provided below.(1)$$Sensitivity=\frac{\left(True\;Positives+False\;Negatives\right)}{True\;Positives}$$

#### 2.7.2. False Positive Rate

The False Positive Rate can be described as the ratio between the false positive results out of the total negative instances, which means that there are negative instances that the system has wrongly identified as positive.(2)$$FPR=\frac{\left(False\;Positives+True\;Negatives\right)}{False\;Positives}$$

#### 2.7.3. Precision

Precision can be described as the ratio between true positive predictions compared to the total predicted positives. It evaluates the correctness of positive predictions. Precision can be low when there are variations in view angles if the system is trained on front-facing frames, but during testing with variations in angles, the model’s precision decreases, and it may incorrectly match subjects from the wrong angle. During variations in clothing, precision drops if the system misidentifies an individual based on clothing appearance. Precision also drops with occlusions, walking speeds, and lighting conditions, and it is only applicable when minimizing false positives, such as in biometric access control or surveillance. The formula is provided below.(3)$$Precision=\frac{True\;Positives}{\left(True\;Positives+False\;Positives\right)}$$

#### 2.7.4. F1-Score

F1-score can be described as the harmonic mean between recall and precision. This metric considers both false positives and false negatives. The method works well with view angles since it measures how the system handles false negatives and false positives, but variations in view angles will cause a noticeable drop in accuracy, which is clearly visible in the F1-Score. In clothing variations, the F1-Score will impact precision and recall, and it is widely used in clothing variation covariance scenarios. However, walking with a backpack or coat will be evaluated as false positives and false negatives. The F1-Score is used in real-time surveillance applications when we want to balance precision and recall, and the system needs to handle covariates like view angle changes, clothing differences, and occlusions. The formula is provided below.(4)$$F1-Score=2\times \frac{\left(Precision\times Recall\right)}{\left(Precision+Recall\right)}$$

#### 2.7.5. Receiver Operating Characteristic

Receiver Operating Characteristic is a visual representation that shows the relationship between the True positive rate against the False Positive rate, with variation in the classification threshold.

#### 2.7.6. Area Under the ROC Curve

AUC quantifies the region that is below the ROC curve, represented as a scalar value ranging between the interval 0 and 1. A higher value of AUC is indicative of good models. The AUC is very useful when there are variations in the view angles, as the method is responsible for evaluating how well the system distinguishes between individuals at various thresholds, and it will distinguish the gait patterns even with variations in the view angles. The AUC method is also sensitive to variations in clothing because it identifies how well the model can distinguish between people, even when they wear different clothes. The AUC is useful in the view of angle variations, clothing, and speed variations, and is effective in biometric identification.

#### 2.7.7. Equal Error Rate (EER)

EER refers to the point where the False Acceptance Rate equals the False Rejection Rate. EER serves as an important biometric measure, as it provides a single value that represents the balance among false acceptances and false rejections. EER works well in a system where a balance between false rejections and false positives is required, but it fails on unbalanced datasets or in cases where the false positive rate is much higher compared to false negatives, which might be the case in security-sensitive applications. EER is useful in different view angles because it maintains a balance between the false positives and false negatives threshold, but the method can still be affected by extreme variations in view angles. EER is sensitive to clothing variations, and in real-time applications, where clothing frequently changes, it helps to show the trade-offs in system performance. EER is useful in covariance scenarios such as view angles, occlusions, and clothing variations, but it is not applicable in low-light conditions and does not perform well in real-time verification tasks.

#### 2.7.8. Cumulative Matching Characteristic (CMC)

The CMC Curve is a plot that depicts the correct match in terms of the likelihood to be in the list of the first k-ranked candidates. The CMC curve is broadly used in gait recognition to evaluate how well the system ranks potential matches in retrieval tasks. CMC is useful in variations in view angles; for example, a top-k match could retrieve the correct person, even if the Rank-1 match fails due to variations in view angles. It also works well with clothing variations, occlusions, and walking speeds. It is useful in multi-shot identification, as it helps assess how well the system ranks the correct person in the top-k positions, which is important in crowded surveillance scenarios.

#### 2.7.9. Rank-1 Accuracy

It is the proportion of test individuals, where the accurate match is identified at the initial position of the list of matches in order. Rank-1 accuracy is relevant for identification tasks, which aim at identifying the subject per the first attempt. Rank-1 accuracy is useful in view angle covariance conditions when a person is camera-facing or at a 90-degree angle. However, when there is a change in the view angle from 90 degrees, the method starts performing worse. The method is also not useful in cases of clothing variations and does not perform well in occlusions since it focuses on the first match and does not account for features that might be obscured or difficult to detect. If there are variations in walking speed, such as walking too fast or too slow, Rank-1 accuracy does not reflect how well the model performs across speed variations. The method is only applicable in controlled environments where conditions, such as clothing, lighting, and view angles, remain constant, and it is not ideal for real-time surveillance applications (see Table 2).

### 2.8. Gait Recognition Under Different Covariates

Covariates are independent variables that reduce the effectiveness of the gait recognition model. These are carrying conditions, speed variations, clothing differences, cross-view angles, occlusions, view variations, and spatial and temporal conditions, as shown in Figure 9. Loose or restrictive clothing, for instance, can seriously influence a person’s movement, which in turn poses a problem as far as recognition is concerned. The use of heavy clothing can also impact a person’s walking style. Moreover, numerous angles and carrying conditions, such as bags, briefcases, and backpacks, can decrease the accuracy of the model.

Different types of covariate factors that affect the accuracy of the gait recognition system are discussed below in detail.

***View Variations:*** View variations refer to the changing of the view angles in which an image is captured. The variation in the view angle affects the accuracy of the gait identification system. The gait identification system depends on whether the person is seen from a front view (90 degrees), side, or rear. Gait datasets contain variations in the view angles, such as the CASIA B dataset, which contains view angles from 0 to 180 degrees, and the OVMULP pose dataset, which contains angles from 0 to 270 degrees.

***Occlusion:*** Occlusion refers to when the subject’s body is fully or partially blocked, which creates issues for the recognition process of gait identification, especially for model-based approaches where the human pose is initially required to verify the gait patterns. Many existing gait datasets may not contain occlusion frames, and there is a need for new gait datasets in the future that contain occluded frames.

***Carrying Conditions:*** Carrying conditions involve the subject holding an object in their hand, such as a hand carry, bag, ball, etc. Carrying conditions can negatively impact the accuracy of the gait recognition system. Some gait datasets, like the CASIA B dataset, include hand-carry conditions, while the GREW dataset features ball-carrying scenarios.

***Speed Variation:*** Speed variations refer to variations in the walking speed of an individual, which can affect the accuracy of the gait biometric system. The CASIA C dataset contains variations in walking speed, such as slow walk, normal walk, fast walk, and normal walk with hand carrying.

***Clothing Variation:*** Clothing variations refer to wearing different types of attire, such as a T-shirt, long coat, cap, etc. These variations in clothing significantly affect the accuracy of the system. The CASIA B, TUM Gait, and OUISIR datasets contain different clothing frames.

***Spatial and Temporal Conditions:*** Spatiotemporal features describe how the environment and timing related to gait sequences affect the accuracy of the system. Variations in lighting conditions or time of day, such as at night, can affect the model’s accuracy.

***Cross View:*** Cross view means the subject is captured from different view angles, such as from the front, side, or rear. The accuracy of the model decreases if the view angles are continuously changed.

Over the past few years, a variety of techniques have been used for gait recognition. Human gait-based techniques are typically classified into two categories: model-based techniques and model-free methods, as depicted in Figure 10.

#### 2.8.1. Model-Based Techniques

In model-based approaches, features are derived based on modeling of the human body structure and analysis of the motions of various body segments. These methods work well under various covariate conditions because they are less sensitive to changes in human appearance related to model-free approaches. The pose estimation model-based pipeline is defined in Figure 11.

In [22], the relationship between gait elements in 3D dimensions and human body movement is established. LSTM and residual networks are utilized for extracting spatial and temporal gait characteristics. For temporal features, the LSTM network consists of 2 LSTM layers with an input size of 88 × 50, along with a Tanh activation function. The residual block has an architecture similar to ResNet50, with a ReLU activation function. The two features are combined, and there is a SoftMax layer for the classification. The CASIA B dataset is adopted for training and evaluation of the proposed model. In the research [37], pose sequences are generated with the help of the Lower–Upper Generative Adversarial Network. In order to learn full-rank matrices from the pose sequences, LUGAN utilizes the adversarial training process. The research uses the plug-and-play technique, in which all convolution blocks are replaced with spatial graph convolution blocks to extract high-level gait features from the pose network. LUGAN consists of GCNs as well as FCNs; the planned technique is trained using CASIA B and OUMVLP-Pose datasets. In the research [38], Pose Gait methods are used to extract information from 14 different human joints. These extracted data are then passed to a CNN network to capture spatiotemporal features. Finally, SoftMax loss and center loss are applied to differentiate between various classes. Pose Gait is introduced to handle clothing variations and carrying conditions. It extracts 3D human poses using the OpenPose technique. The spatiotemporal feature is extracted using the CNN technique, such as human joint angles, limb lengths, and joint motion. These features are then fused at the end for comprehensive analysis.

In the research [39], pose estimation maps are extracted for each body part from the input frames by processing pose sequences. These pose maps preserve global and local information from the input frames, which are less affected by noise and covariate factors. The pose estimation maps are generated using a CNN model called AlphaPose. These maps for each body part are then aggregated into a single heatmap for each frame, which captures global information while reducing redundancy. The resulting heatmaps are passed to a CNN network known as GaitMap-CNN, which extracts relevant frames responsible for capturing changes in the structure of the human body over time. Simultaneously, a Graph Convolutional Network (GCN), known as GaitPose-GCN, captures movement information of the body joints. This network creates a graph that associates the joints of the human body with each frame, enabling the system to track the body’s movement more effectively. In the final stage, both the heatmap features and the pose graph are combined to form the final gait feature. These features are instrumental in identifying an individual based on their walking style. The model is trained using a triplet loss function, which helps optimize the learning of these gait features. In the research [40], to address covariate factors such as clothing variation, view angles, and occlusion, a fusion of different methods is employed. First, 3D human poses are extracted from the input frames using body parsing and virtual dressing techniques. Body parsing is used in gait recognition to segment the human body into different regions to extract meaningful information from specific body parts. Virtual dressing involves applying digital clothes to the 3D model to analyze how different clothes affect the accuracy of the gait model. Using body parsing and virtual dressing, 3D body parameters are generated. These parameters are then transformed into a 2D structured matrix through gait semantic folding, which simplifies the data processing. The model learns gait patterns with respect to time using a Hierarchical Temporal Memory (HTM) network. To enhance the performance of the model under various covariate conditions, a top–down attention mechanism is introduced. Finally, the learned gait patterns are converted into Sparse Distribution Representations, ensuring accurate gait recognition.

In the research [41], a Graph Convolutional Neural Network and 3D pose estimation techniques are used to handle covariate factors such as clothing variation and different view angles. The gait feature extraction method is divided into two steps: intra-frame features and inter-frame features. In the intra-frame features, human joint angles and positions are extracted, while the inter-frame features capture the dynamic movements of the joints. The Graph Convolutional Neural Network processes the gait data, where each joint of the human body is treated as a node, and the relationships between the joints are represented as edges in the graph. This helps the model learn both spatial and temporal gait features. A Siamese network is used to measure the similarity between different samples with the help of a Graph Neural Network-based matching module, which updates the nodes and edge information to ensure more accurate gait pattern matching. To differentiate between individuals based on gait patterns, Supervised Contrastive Loss is employed.

In research [42], skeleton-based techniques are employed to improve the accuracy of the gait recognition system. The process starts with centroid-based skeleton alignment, which makes sure that the human skeleton is centrally aligned in terms of scale and direction. This alignment is based on the torso joint centroids, which remain unaffected by human movement. Next, a dual-stage linear matching process is introduced to match the input frames with the registered frames. A cost matrix is used to calculate the similarity between the frames, and the matrix is adjusted according to the input skeleton. This adjustment helps reduce the chances of noisy frames being matched incorrectly. Finally, a weighted majority voting scheme is applied to distinguish between frames. The frames with the best quality are assigned more weight, enhancing the accuracy of the gait recognition system. In the research [43], skeleton-based methods are combined with spatial–temporal graph convolutional networks and Canonical Polyadic Decomposition to improve gait identification system performance, which is affected by various covariate factors. Skeleton information from input frames is extracted using a CNN-based method, such as OpenPose. This information includes joint data that represent body parts in the input frames. The convolution operation is applied to these joints using ST-GCN, where each joint of the human body is treated as a node in the graph network. Spatiotemporal features are then extracted with the combination of spatial graph convolutions and temporal convolutional networks. The ST-GCNs capture both local and global body patterns, such as arm swinging or leg bending, which are crucial for gait recognition. After gait feature extraction via ST-GCNs, the features are further optimized using Canonical Polyadic Decomposition (CPD). This step eliminates redundant and less useful features, retaining only the important features that improve the robustness and accuracy of the gait recognition system. In the research [44], the MS-Gait technique is introduced, which combines graph neural networks with skeleton data to accurately identify humans under different covariate conditions. The technique focuses primarily on bone and motion information to distinguish between different gait patterns. The graph convolution network creates spatiotemporal relationships among human body joints, treating the human skeleton as a graph, where each joint is a node. The multi-stream technique processes three different features: joint, bone, and motion data. The bone-related data represent the length and direction between joints, while the motion data capture the speed and movement of different body joints. The multi-stream feature extraction block increases the performance of the gait identification system by capturing both static and dynamic gait patterns. To further improve the feature selection process, Squeeze-and-Excitation (SE) blocks are integrated with the graph convolutional network. These blocks reweigh feature channels based on their importance, optimizing the model’s ability to focus on the most relevant features.

In the research [45], GaitGraph is combined with 2D human pose estimation to accurately identify human gait. The graph convolutional neural network, along with the 2D pose estimation model, extracts more robust gait features compared to traditional silhouette-based techniques. GaitGraph takes the human skeleton as input, effectively addressing challenges like clothing variations and carrying objects. The 2D pose estimation extracts information from key joints, where the nodes of the graph represent body joints, and the edges represent the bones connecting these joints. The ResGCN blocks in the graph combine graph convolution with 2D convolution, establishing spatial and temporal relationships between the joints. ResGCN consists of bottleneck layers and residual connections, which increase the learning capability of the network. The temporal features are processed using sequence-based processing to maintain the temporal dynamics of the gait. In the research [12], humans are detected using the pretrained YOLOv2 model, which is trained on the COCO dataset. Spatial features are related to the location, while temporal features are associated with the time at which the features are extracted. CNN models are then employed to extract human pose features, resulting in 16 key joints. This joint information is passed to the LSTM for temporal gait feature extraction. An SVM classifier is subsequently used to classify the individuals. The accuracy of the model is relatively low due to the limited size and quality of the training dataset. The model is trained on a custom dataset, with frames captured from 11 different view angles, ranging from 0° to 180°. In the research [46], stacked autoencoders are utilized to derive gait features from the 3D human skeleton, which work well under different covariate conditions and view angles. The extracted features are encoded and passed through biLSTM to obtain more discriminative gait features. The primary goal of using autoencoders is to convert 3D skeleton joint coordinates into a standardized canonical side view (90°), which enhances the robustness of the gait features. The autoencoder consists of three stacked layers, with different layers handling different features. The first layer handles small pose variations from angles 0° to 36°, the middle layer addresses larger pose variations from angles 36° to 72°, and the final layer converts the pose information into the side view at 90°. The stacked autoencoders are trained using a greedy layer-wise algorithm, where every encoder is trained independently. A multi-loss strategy, combining SoftMax loss and center loss, is used to improve the accuracy of the extracted gait features. The CASIA A and CASIA B datasets are used for training and testing the model.

In the research [47], pretrained models MobileNetV1 and Xception are used to extract gait features from gait images. During the feature extraction process, the fully connected layers of both models are removed to focus on the most relevant gait features. MobileNetV1 consists of depthwise convolutional layers, followed by pointwise convolution with an output vector size of 1024. In contrast, Xception only utilizes separable convolutional layers, along with global average pooling, resulting in an output vector size of 2048. The next step involves concatenating features from both deep learning models, resulting in an output vector size of 3072. To reduce the feature dimensions, PCA is applied, which compresses the feature vector size to 620. For the classification task, both Random Forest and SVM classifiers are employed, with SVM utilizing the one-vs-all classification method. Encoders are becoming increasingly popular in the field of computer science, especially in gait recognition, for the purpose of feature extraction. In the study [48], stacked autoencoders are utilized for this purpose. Initially, the input image is resized to 28 × 28 pixels and converted to grayscale. Two autoencoders are employed together, with the first autoencoder trained using 250 hidden layers for 100 epochs, and the second autoencoder also consisting of 250 hidden layers trained for 100 epochs. ReLU is used as the activation function during training to mitigate the issue of vanishing gradients. Once autoencoders are trained, the encoder parts of the autoencoders are used to extract features, which are then passed to classifiers for the classification task. Different classifiers, including SVM and KNN, are employed. The model is trained and tested on the CASIA B dataset. In the study [49], a Graph Convolutional Neural Network is used to extract spatial and temporal features from input frames. The pretrained OpenPose network is used to extract information about 25 key joints from the frames. These joints represent the human body’s skeletal structure and are utilized to track the human walking cycle. The human skeleton information is represented as an undirected acyclic graph, where humans are depicted as vertices, and the edges represent the relationships between the joints. The GCNN is used to extract both spatial and temporal information from the body joints. Residual connections within the graph network are employed to capture lower-level features from the shallower layers. To reduce the number of layers, the Global Attention Sum Pooling method is applied, focusing only on the most significant gait features. The output of the Global Attention Sum Pooling is a 1D vector, which is used for classification. For loss functions, Categorical Cross-Entropy Loss and Triplet Loss are utilized. The model was trained on the CASIA B dataset by utilizing RMSProp optimizer with a batch size of 64, and the drop rate is 25%. In the research [36], the motion of humans is captured to generate a codebook with the help of Fisher vector encoding, and linear SVM is used for classification. The model is trained and tested using TUM Gait and CASIA A datasets (see Table 3).

#### 2.8.2. Model-Free Techniques

Model-free techniques are commonly known as appearance-based or holistic-based approaches because model-free approaches focus on the shape of the human body and its motion. Model-free techniques derive gait characteristics straight from the human silhouette and typically involve several preprocessing steps, such as background subtraction, foreground subtraction, normalization, feature extraction, and classification. One benefit of model-free methods is that they do not rely on high-resolution images and can work well with low-resolution pictures, making them less computationally expensive [50]. The gait motion of a walking human is represented in a compact form through its appearance over time, without considering any underlying body structure. This is a holistic approach, where the derived methods are not tied to a specific object. Interestingly, the methods used for human gait detection can also be applied to animal gait with some minor modifications. Model-free methods are considered amongst the most effective methods for gait recognition. The model-free methods process is defined in Figure 12.

For gait recognition, a lightweight model is used, combining two pretrained deep learning models: VGG19 and MobileNet-V2. The deep learning model is fine-tuned to handle various covariates in order to improve the accuracy and computational time of the model using the TUM Gait and CASIA B datasets. Both models are trained through transfer learning, without freezing any layers, resulting in newly trained models. Feature engineering for VGG19 is performed on the last layer, while for MobileNet-V2, it is conducted on the last pooling layer. The extracted features are then combined by using Discriminant Canonical Correlation Analysis. The process of feature extraction is carried out with the help of global average pooling layers, which are combined using the DCCA method. To optimize the feature extraction process and select only the most useful features, an improved version of Moth-Flame Optimization is applied. For the classification process, an Extreme Learning Machine algorithm is used, which improves the accuracy of the model and its computational time [14]. In the study [51], the model is provided with Gait Energy Images (GEIs) as input to address covariate factors such as normal walking, clothing variations, and walking while carrying objects in a Multiview environment. The GEI images are preprocessed using random sampling, ensuring that individual images remain unaffected. For feature extraction, Histogram of Oriented Gradients (HOG) is applied to capture gait patterns by calculating the gradient magnitudes and orientations in localized image regions. These gait patterns are then passed to a SVM for classification, and the model is trained on the CASIA B dataset. A limitation of the model is its insufficient focus on view angles. Additionally, for unknown covariates that are not present in the datasets, methods are needed to address these factors, as they can negatively impact the accuracy of gait recognition models.

To address the issue of covariates, two models are proposed. The first model is a CNN designed to handle known covariates, while the second model is a classification method based on discriminative features to handle unknown covariate conditions. The CNN model uses Gait Energy Images, which are processed through CNN to address known covariates. The CNN consists of 4 convolutional layers with a 3 × 3 filter size and no padding, along with Leaky ReLU as the activation function. After each convolutional layer, a max pooling layer with a 2 × 2 window size is applied. Following the convolution and pooling layers, there are fully connected layers with an n-dimensional output, and a SoftMax layer is used as the final output layer. The second approach focuses on searching and extracting covariate-invariant features, derived from both the gallery and probe sequences. For feature extraction of unknown covariates, methods like Local Binary Patterns, Histogram of Oriented Gradients, and Haralick texture feature descriptors are used. To reduce dimensionality and select relevant features, Fisher Linear Discriminant Analysis is applied. The covariate factors addressed in this approach include clothing conditions and walking speed [52].

A simple, lightweight CNN model is designed to handle various types of variations, with occlusions being less influential on the proposed method, which typically reduces the accuracy of gait recognition. The model takes Gait grayscale images as input at the size 240 × 240, which are processed through several convolutional layers. The model consists of 4 convolutional layers: the initial layer has 16 filters of size 3 × 3, the second layer has 32 filters of size 3 × 3, the third layer has 64 filters of size 3 × 4, and the fourth layer has 124 filters of size 3 × 3, with a ReLU activation function. Max pooling layers with a 2 × 2 window size and a stride of 2 are applied to reduce the dimensions of the convolutional layers. After the convolution and pooling layers, there are fully connected layers with 1024 neurons, followed by a SoftMax layer. The model is trained using the Adam optimizer for 50 epochs, and it addresses only a few covariates, specifically, those found in the CASIA B dataset, which consists of images captured from a single view angle (90°) [53]. In the study [54], the process is divided into two phases. The first phase focuses on classifying human gait from video frames using a convolutional bidirectional LSTM (BiLSTM) network. In the second phase, deep features are extracted with the help of the FireConcat-02 layer in the YOLOv2 SqueezeNet model and then passed for human gait recognition and localization using the TinyYOLOv2 model, alongside the predicted scores. For the first phase, ResNet-18 is applied for initial feature extraction, while LSTM is used to extract temporal features. The BiLSTM has 1000 hidden units and a dropout rate of 0.5 to prevent overfitting. The initial spatial features, extracted by ResNet-18, are passed to the LSTM to extract temporal features. In the final step, a SoftMax activation is used to classify the individual. This research focuses on clothing covariates, walking speed, and view angles, and the CASIA dataset is employed for both training and testing. A gait recognition model is designed to handle occlusions and different clothing conditions using two deep learning models. CNNs are employed to extract spatial features of pedestrians, while LSTM is used for extracting temporal features from the training videos. The input image has a size of 224 × 224. The first convolutional layer consists of 64 filters of size 7 × 7 with a stride of 2. Following this, there are 4 residual blocks: the first residual block consists of 2 layers of 3 × 3 convolutions with 64 filters, the second block follows the same structure but uses 128 filters, the third block consists of 256 filters, and the last block has 512 filters. In each block, vanishing gradient techniques are applied to avoid the vanishing gradient problem. After these blocks, there are 1 × 1 convolutional layers, along with fully connected layers containing 512 units. The base model for extracting temporal features is ResNet-34. By stacking more layers, the model can handle more complicated tasks. Since CNNs are not capable of extracting time-based features, and the features extracted are not spatiotemporal, LSTM with 100 units in each layer is incorporated. The model has 3 LSTM layers to capture spatiotemporal features, and SoftMax is used for classification. For model training and evaluation, the CMU MoBo dataset is used, which includes clothing covariates, backpack walking, and ball holding [55].

The process of person identification across different cameras is a critical task in computer vision. In this research, person re-identification is performed using a model-free technique. The method begins with angle estimation of the gait. Gait Energy Images are employed to calculate the angle based on the human silhouette. Gait energy images are extracted by using background modeling, initially starting with moving object detection, followed by the application of background subtraction techniques. For angle estimation, the CNN network takes a 224 × 224 image as input. The starting convolutional layer consists of 64 filters of size 7 × 7 with a stride of 2, along with a ReLU activation function. The second convolutional layer of the model consists of 128 filters of size 3 × 3 with a ReLU activation function, and the third convolutional layer consists of 256 filters of size 3 × 3 with a stride of 1, also using a ReLU activation function. After each convolutional layer, a max pooling layer with a 2 × 2 kernel and a stride of 2 is applied. At the end, a fully connected layer with 512 neurons is used, followed by a SoftMax layer for classification. For gait recognition, the model consists of three convolutional layers, with filter sizes of 64, 128 and 256 and strides of each filter 2, 1, and 1, respectively, using a ReLU activation function. The final layer is a SoftMax layer for classification. This research specifically addresses the view angle issues in different datasets, including CASIA B, OU-ISIR, and OU-MVLP [56]. In the study [57], three pretrained models are employed for gait recognition: DenseNet 201, VGG16, and Vision Transformer. Gait Energy Images (GEIs) are extracted by averaging the human silhouette, and these images are then passed through the pretrained and fine-tuned models. The initial layers of DenseNet 201 consist of 7 × 7 convolutional layers with a filter size of 64 and a stride of 2, followed by a 3 × 3 max pooling layer with a stride of 2. DenseNet 201 has 4 dense blocks, with each block consisting of bottleneck layers of 1 × 1 convolutional layers, with batch normalization and a ReLU activation function. Transition layers are interspersed between the dense blocks, which reduces the dimensions of the feature maps. The final layers of the model include fully connected layers followed by a classifier layer, which generates the final output. VGG16 consists of 13 convolutional layers with a filter size of 3 × 3, and there is a fully connected layer for classification. Additionally, the Vision Transformer divides the image into patches, which allows the model to focus on important gait features in specific areas of the image. All models generate scores for the output, which are aggregated to produce the final prediction. The datasets utilized in this study include the CASIA B dataset, the OU-ISIR dataset D, and the OU-LP dataset. In the study [58], an attention-based method called Vision Transformer (ViT) is employed for gait recognition. Gait Energy Images (GEIs) are divided into smaller sections of size 32 × 32 and converted into sequences through flattening and patch embedding. The ViT model uses attention mechanisms such as multi-head self-attention to focus on important regions present in the image for better recognition. To restore the positional information of the patches, position embedding is used alongside the patch embedding and is applied to the patch sequences. After that, the vector series is forwarded to the Transformer encoder, which generates a gait vector representation. The encoded patches are then classified using multi-layer perception, which predicts the class label. The model is trained on three different datasets: CASIA B, OU-ISIR, and OU-LP. The model achieves high accuracy rates because it effectively handles noise and incomplete silhouettes.

Due to the limitations of handcrafted models, the study [59] uses a ten-layer convolutional neural network for feature extraction, with GEI as input. The model consists of 10 CNN layers, including 4 convolutional layers, where each convolutional layer has a filter size of 3 × 3, along with a ReLU activation function. To reduce the dimensions, a pooling layer follows each convolutional layer. The final layer consists of a fully connected layer with 1024 neurons, and a dropout value is set to 0.5 to prevent overfitting, followed by a SoftMax layer for gait pattern classification. The model is trained using the CASIA B dataset. In the study [60], a manual attention technique is employed to guide the training process and extract crucial gait features. The model learns less when the silhouette is the input, as it limits the model’s learning. To handle this problem, an attention model called AttenGait is introduced. Initially, the model extracts important joint features such as limb information from optical flow maps using convolutional layers. Unlike traditional GEI, AttenGait contains broader gait feature information, which helps overcome the constraints of silhouette-based methods and handles various covariate factors. AttenGait consists of three blocks. The first block, called Attention Conv, focuses on important regions beyond temporal features. The second block, Spatial Attention HPP, captures features from the frames by applying horizontal crops. The final block, known as Spatial Attention HPP, extracts temporal features from the human joint. The spatial and temporal features are integrated in the attention block, and K-Nearest Neighbors (KNN) is used for classification. The CASIA B and GREW datasets are employed for training and testing the model. In the study [61], Generative Adversarial Networks (GANs) are used to generate human images with varying covariates, such as carrying objects and clothing variations, based on the CASIA B dataset. Pretrained models, including AlexNet, Inception, VGG16, VGG19, ResNet, and Xception, are trained on the frames generated by the GAN. To address class imbalance, the Synthetic Minority Over-sampling Technique (SMOTE) is applied. The best gait features are selected by using Particle Swarm Optimization and Grey Wolf Optimization techniques, which help identify the most relevant gait patterns extracted from the input frames. CNN classifiers are then employed to perform the classification task. In the study [62], a deep multi-layered convolutional stacked capsule network is used to improve the quality of gait energy images and handle covariate factors such as clothing variation, view angle, and carrying objects. The capsule network uses a series of convolutional layers to extract high gait patterns, which include the color and type of clothing. The function of Capsule networks is to extract hierarchical and spatial relationships between features. These features are extracted using primary capsules, which encode the spatial features and improve the gait recognition process. Within the capsules, a dynamic routing algorithm is employed to adjust the connections and transfer information more accurately.

In the study [63], a fusion technique combining convolutional networks and a Bayesian model is employed. In the initial phase, human optical flow-based motion is estimated using the Horn–Schunck technique. This process calculates motion between two input frames by utilizing optical flow estimation. The deep learning model EfficientNet-B0 is fine-tuned on motion frames to improve model performance, and Bayesian optimization is applied to dynamically optimize hyperparameters of the model rather than relying on static parameters. The internal structure of EfficientNet-B0 consists of 3 × 3 convolutional layers with 32 filters, along with MBConv blocks that include depthwise separable convolutions, pointwise convolutions, and residual connections. Other components of the model include a global pooling layer, a fully connected layer, and a SoftMax activation function in the output layer, with Swish activation applied throughout the network. A video enhancement method known as sequential contrast is used to improve the visibility of gait patterns within the input frames. The Sq-Parallel Fusion technique is employed to combine motion-based and enhanced video methods, and the most relevant features are selected using Entropy-controlled Tiger Optimization (EVcTO). To classify the gait features, an Extreme Learning Machine (ELM) classifier is utilized. The model is trained and tested on the CASIA B and CASIA-C datasets. The study [64] employs 3D reconstruction of the human body to filter out irrelevant information in gait recognition. Rather than relying on human silhouettes or skeletons, the researcher uses 3D human body reconstruction to preserve key gait features within RGB frames. Initially, the Human Mesh Recovery method was used to generate a 3D human body mesh from the input RGB frames. The method takes human pose and shape information as input, which provides a discriminative representation of gait. To extract frame-level features, the frame-level part feature extractor is introduced, consisting of three convolutional blocks. Each block divides the feature map into smaller sections and performs convolutional operations on each part. The outputs of the blocks are then combined to obtain more refined gait features. The Multi-Granular Feature Fusion block extracts both spatial and temporal gait features and is composed of three blocks. The initial block of the model is responsible for capturing global spatiotemporal features, while the second and third blocks focus on extracting body part-level features. The Micro-Motion Template Builder, part of the Multi-Granular Feature Fusion block, extracts macro-level features from body part-level features. This block uses 1D convolutional layers along with a channel attention mechanism to capture small gait features. The model is trained using a triplet loss function to optimize the embedding space. In the study [8], spatiotemporal features are extracted using a novel technique called STAR, which stands for Spatio-Temporal Augmented Relation Network. The model takes gait silhouette input and represents it as a 4D tensor. The backbone block of the model generates feature maps, and the block consists of three 3 × 3 convolutional layers with a stride of 2, using filter sizes of 32, 64, and 128 filters, along with pooling layers. The Multi-Branch Diverse-Region Feature Generator includes multiple branches, each designed to extract gait features from different human body parts. The initial branches focus on the upper portion of the body, like the head and torso, while the lower branches target the lower body parts, such as the legs. The features from these different branches are then combined. The Spatio-Temporal Augmented Interactor integrates intra-relationships and inter-relationships. Intra-relationships use spatial attention techniques to capture the relationships among different body parts, while inter-relationships use temporal attention mechanisms and 3D convolutions to capture temporal relationships across different frames. The Micro-Motion Template Builder uses 1D convolution to collect micro-level characteristics. Finally, the outputs of the Spatio-Temporal Augmented Interactor and Micro-Motion Template Builder are passed to fully connected layers to aggregate the features. These aggregated features are then passed to the classification layer with a SoftMax activation function for classification (see Table 4).

#### 2.8.3. Fusion Model

Fusion-based techniques combined both model-based and model-free techniques in gait recognition. The model is capable of capturing human silhouette features as well as skeleton information from the gait. These models perform better under different covariate conditions.

Long-term person re-identification is a critical task in gait recognition systems, especially for tracking specific targets in surveillance videos. In the study [65], a two-branch person re-identification model is proposed, which integrates both appearance and gait information. The study uses an optimized Sobel Masking Active Energy Image instead of traditional Gait Energy Images, which preserves gait information more effectively. For appearance-based feature extraction, ResNet-50 is employed, while gait features are derived using the Improved-Sobel-Masking Active Energy Image. This method effectively removes irrelevant details from the frames and only preserves the key features related to the gait cycle. The Improved Sobel Masking Active Energy Image identifies dynamic regions of the frames, such as limbs, by calculating the difference between input sequences and removing static body parts like the torso. These processed frames generate an Active Energy Image that captures the full gait cycle. After this, Sobel edge detection refines these gait patterns by finding the dynamic regions, and wavelet de-noising is applied to remove noise from these regions. In the end, the Improved Sobel Masking technique is applied, combining Sobel edge detection with wavelet-based de-noising to remove the mid-body regions, which are less significant. A fusion strategy is then applied to combine both appearance and gait features, which enhance the accuracy of the re-identification process. The datasets used in this study are the TUM Gait and CASIA B datasets. In the study [66], Skeleton Gait Energy Images (SGEIs) and Gait Energy Images (GEIs) are used as input to a multi-branch Convolutional Neural Network (CNN), which extracts features from both types of input images. The SGEIs are combined with GEIs to enhance the extraction of gait features and to effectively handle challenges such as view angle changes and clothing variations. The dual-stage CNN network is used to extract skeleton joint information. In the first stage of the CNN, confidence maps are generated to identify the body parts, while the second stage generates part affinity fields that represent the relationships between different body parts. The SGEIs are created from skeleton keypoints, and these keypoints are expanded into specific widths and sizes to form a human model. The Hybrid Descriptor combines both GEIs and SGEIs to improve gait recognition. To address covariate factors, a special block called the View Transformation Model is introduced. This model decomposes gait characteristics into subject-independent and view-independent matrices. Similarity is calculated using Euclidean distance, combining the distances for both GEIs and SGEIs.

In the research [67], fusion-based techniques are used for gait recognition. The technique is divided into two phases. The initial phase follows a model-free approach, where human silhouettes are extracted from RGB images, and gait energy images are obtained by averaging the human silhouettes. Gait features from the gait energy images are collected using a CNN network, which consists of a series of convolutional layers with a small number of filters (32) and a kernel size of 32, followed by a max pooling layer and a dropout layer. The second phase employs a model-based technique, where a pose estimation technique is used to extract human pose sequences. Different human joint information is obtained from these pose sequences using the Mediapipe algorithm. The pose sequences contain static features, such as the distance between different body joints, like the distance between the left and right knees. These static features provide spatial information, while dynamic features, such as the knee angle while walking, provide angular information of the body joints. The features from both models are fused into a single feature vector, which contains rich information about the gait sequence. For classification, the concatenated feature vector is passed through fully connected layers to recognize an individual. The model consists of various fully connected layers, with the final layer being the output layer using a SoftMax classifier. The model is trained on the CASIA dataset. The model effectively handles different covariate conditions such as clothing variations, view angles, and walking speeds. It demonstrates excellent accuracy in recognizing individuals under clothing variations and varying view angles, but the computational time of the model increases as a result. In the research [68], a fusion method is utilized in which model-free approaches are combined with model-based methods. The main aim of combining both techniques is to increase the model’s performance under different covariate conditions. In the model-free approach, the GaitGL method is used to extract the human silhouette from RGB images. Initially, a background subtraction method is used to create the human silhouette, and the quality of the silhouette is improved using histogram equalization. After this, various morphological operations are applied to remove gaps in the human silhouette, and normalization techniques are used to increase the resolution of the silhouette. The CNN network for feature extraction consists of convolutional, pooling, and fully connected layers, which produce an output vector. In the model-based technique, a human pose is extracted using the HRNet algorithm, which contains different information about body joints. The pose sequences contain static features and dynamic features. The static features include the distance between the joints, while the dynamic features contain temporal information. A graph convolutional network is utilized for extracting gait features from the pose sequences. In the GCN, each joint of the body part is represented as a node, and the edges connect adjacent joints. The GCN captures spatial information from the body joints, enabling the model to learn complex features. The model-based and model-free techniques are combined by using Compact Bilinear Pooling, which integrates the rich human silhouette and pose estimation features. In the end, the feature vector is passed to the fully connected layer with a SoftMax layer for classification. The model is trained on the SOTON-small and CASIA datasets with triplet and Cross-Entropy Loss functions (see Table 5).

Various other approaches have also been highlighted in the literature, offering contributions to gait recognition systems, either directly or indirectly, under different covariate factors [8,69,70,71,72,73,74,75,76,77,78,79].

## 3. Results

Over the last few years, the research community has put forward several remarkable methodologies for the gait recognition system. Numerous approaches and methods have been employed by implementing several modifications and have been employed to attain topmost accuracy by utilizing various datasets. State-of-the-art techniques have been used to address covariate factors and view angles, thereby improving gait recognition accuracy. Model-based as well as model-free techniques have been used, along with the fusion of both model-based and model-free techniques utilized to obtain better results. Table 2 presents a summary of the prior studies for comparison and analysis.

Some studies have reported outstanding results; however, the majority of the findings are based on less challenging datasets. There is often a lack of variability in lighting, viewing angles, and clothing conditions in the experimental data. Furthermore, a few number of datasets, such as CASIA, TUM Gait, and OU-ISIR, include covariate conditions. These challenges are often either ignored entirely or only partially considered in the research to date [51,52,54,56,57]. In gait recognition systems, experiments usually involve the use of high-resolution images, but factors such as low-light images, different weather conditions, and occlusion images are rarely addressed. Low-quality images make detection difficult due to noise residuals and artifacts, which might affect the performance of recognition systems.

It can be seen from the literature that most of the research has been tested on datasets that include covariate factors. Few studies place emphasis on clothing conditions but fail to address view angles. Other limitations in the research include a lack of focus on low-light conditions and low-resolution images, and, for model-based approaches, pose estimation is only performed when the person is facing the camera. It has also been noted the fact that models can work good on datasets with a small number of covariate factors, and as the quantity of covariates increases, the effectiveness of the models tends to degrade. The models are not tested in real-world environments. In most datasets, the images or videos are of high resolution, but in actual gait recognition scenarios, there are varying lighting conditions, different human poses, diverse clothing styles, and different types of weather and temperatures. Furthermore, issues like occlusion, background clutter, and subject variability in gait patterns can complicate accurate recognition (see Table 6).

### 3.1. Comparison with Traditional Baselines

To highlight the advancements of computer vision-based deep learning gait identification methods, we compare traditional machine learning methods with advanced CNN-based models in Table 3. Traditional machine learning methods are heavily dependent on handcrafted features. For dimensionality reduction, they depend on PCA and LDA, and for the classification task, methods such as k-NN, SVM, or unsupervised networks like the Kohonen SOM are used. The traditional machine learning techniques achieve moderate to high accuracy only in controlled environments, where few angles, such as the front view, are used with minimal clothing variations and normal walking speeds. When complex covariates are introduced to traditional gait methods, their accuracy significantly drops, such as with multi-view angles, diverse walking speeds, and different clothing variations. In contrast, deep learning techniques such as MobileNet-V2, VGG19, DenseNet, and LSTM can perform better under diverse covariate scenarios and learn spatial and temporal features from the input gait sequences. The performance comparison of traditional methods with advanced deep learning methods is described in Table 7.

The table shows that the PCA + LDA or KSOM perform well on only one covariate in a controlled environment, but the accuracy of the model decreases in real-world scenarios. Meanwhile, deep learning models are trained on large gait datasets and perform well in real-world scenarios.

In gait identification systems, there is often inconsistency in the performance of the models. To understand the reasons behind this inconsistency, a discussion is provided below, which outlines the potential causes and future benchmarks for equitable comparisons.

### 3.2. Inconsistencies in Prior Findings and the Need for Standardized Benchmarks

#### 3.2.1. Experimental Circumstances Impacting Performance Comparisons

Experimental circumstances, including environmental factors such as varying lighting conditions, weather conditions, and background clutter, along with dataset-specific limitations like view angles, walking speeds, clothing variations, and carrying conditions, significantly affect the gait recognition model accuracy. Most gait datasets, such as the CASIA dataset, cover very few covariate factors and are created in controlled environments. For example, the CASIA dataset includes only three covariates: normal walking, walking with a hand carry, and walking with a long coat, with view angles ranging from 0 to 180 degrees. Similarly, other datasets, like TUM Gait, also have limited view angles and covariates, and are designed in controlled settings. When these models, trained on such datasets, are applied to real-world applications, they face challenges. In the real world, people wear diverse clothing, ranging from heavy to light attire, and encounter various lighting and carrying conditions that are not present in controlled environments.

#### 3.2.2. Inconsistent Accuracy Rates

Inconsistencies in accuracy are commonly observed in gait recognition techniques due to factors like low-resolution images, clothing variations, pose estimation challenges, and occlusion. The model performs well on controlled environment datasets like CASIA and TUM Gait, as these datasets lack variations in lighting and occlusions and only address a limited range of view angles. However, the performance of models decreases when applied to datasets or real-world environments with diverse view angles or challenging environmental conditions. Models trained in high-resolution image datasets also struggle when applied in real-time, challenging environments with low lighting or when a person wears heavy clothing.

#### 3.2.3. Future Benchmarks for Equitable Comparisons

Future benchmarks for gait recognition should include a wider range of covariates, such as variation in lighting conditions, low-quality images, occlusion, diverse walking speeds, and different clothing variations in datasets. Additionally, frames should be captured in uncontrolled environments incorporating variations in weather conditions, age, and background clutter. By including these features in the dataset models, they will be better equipped to perform effectively in real-time applications, where diverse covariates and environmental challenges are commonly encountered.

### 3.3. Quantitative Covariate Impact and Model Robustness Comparison

Covariate factors such as clothing variations, view angles, walking speed, and lighting conditions significantly affect the accuracy of gait recognition systems. In this section, we analyze how the model performance is affected when complicated covariate factors are introduced, based on different studies. In study [81], both clothing variation and walking speed have a noticeable impact on model performance. The model’s accuracy was 90.33% when the subject wore normal clothing, but this dropped to 87% when the individual wore heavy coats. Walking speed also plays a critical role; the model performed at 99% accuracy with a normal walking speed, but the accuracy decreased to 93% when walking fast and to 94% when walking slowly.

In study [38], various covariates were considered, including normal walking without any carrying conditions, walking while carrying a backpack, and walking with a hand carry. The model’s accuracy was 63% for normal walking but decreased significantly when carrying items: 42% when walking with a backpack and 31% when carrying a hand carry. In study [82], the model’s accuracy was 98% during normal walking. However, the accuracy dropped to 93% when walking with a bag and further decreased to 80% when carrying a hand carry. In study [83], the model was trained and tested using the TUM Gait dataset. The model achieved 82% accuracy during normal walking. However, when covariates were introduced, the accuracy decreased to 68% when the subject was carrying a backpack and to 76% when wearing coated shoes.

In study [84], the model achieved 99% accuracy with normal walking. However, when the view angles were changed, the accuracy began to decrease. On the CASIA B dataset, the average accuracy across angles from 0° to 180° was 74%, and on the OU-MVLP dataset, the accuracy dropped further to 57% for angles ranging from 0° to 255°. In study [52], the model’s accuracy with normal walking on the CASIA B dataset was 97%. However, when the subject wore a coat, the accuracy decreased to 95%. With the addition of a bag, accuracy further decreased to 91%. When different walking speeds were introduced, the accuracy dropped from 90% with normal walking speed to 83% with fast walking speed and to 85% with slow walking speed. On the ISIR dataset, the model achieved 100% accuracy without any clothing variations. However, when different clothing variations, such as regular pants, full shirts, long coats, mufflers, and short shirts, were introduced, the accuracy decreased to 67%.

In study [85], various covariates were introduced, including carrying a single shoulder bag, backpack, handbag, wearing a coat, and carrying nothing. The model achieved 89% accuracy during normal walking or when carrying nothing. However, the accuracy decreased when carrying different items: it dropped to 79% with a shoulder bag, further decreased to 73% with a backpack, and continued to decline to 65% when carrying a handbag. The accuracy with a coat was 73%.

In the study [53], two gait datasets, CASIA B and CASIA C, are used for model training and testing. On the CASIA B dataset, the accuracy of the CNN-based model is 98% for normal walking. However, for covariate factors such as walking with a bag, the accuracy decreases to 81%, and for walking with a coat, it drops to 92%. On the CASIA C dataset, the model achieves 99% accuracy for normal walking, but when walking with a bag, the accuracy decreases to 88%, and for slow walking, it drops to 95%.

In the study [83], a custom kernel method is used to handle covariate conditions. The method employs three custom kernels: the first kernel extracts static targets from the input frames, the second kernel focuses on dynamic regions, and the third kernel targets dynamic body parts. The model is trained on three datasets: CASIA A, CASIA B, and CASIA C. The accuracy of the model on normal walking without any covariates is 95%. However, when walking with bags, the accuracy decreases to 83%, and when walking with coats, the accuracy drops further to 58%.

In the study [86], the Cross-Covariate Causal Intervention technique is employed to handle covariate factors. The technique uses Causal Intervention Metric Learning to eliminate spurious correlations introduced by various covariate factors. Positive-Guided Negative Selection is applied to select negative samples based on visual similarities. The model is tested on different datasets, including Gait3D and GREW. The accuracy of the model without covariates is 89%, but when covariates are introduced, the accuracy decreases to 69% on Gait3D and to 82% on GREW. On the CASIA B dataset, the model achieves 98% accuracy on normal walking, but this decreases to 89% when clothing variations are included (see Table 8).

Computer vision methods used in gait identification systems still face challenges due to covariate factors such as occlusion, clothing variation, walking speeds, and diverse multi-view scenarios. These challenges arise because datasets used are often not created to cover all covariate factors. Most gait datasets only address a limited number of covariates in controlled environments, where lighting conditions are fixed, there are only two–three clothing variations, and minimal view angles are considered. However, when these methods are deployed in real-world environments, which consist of variations in lighting, adverse weather conditions, clothing types across different regions, and diverse view angles, their accuracy tends to degrade. Additionally, in 2D silhouette or pose estimation, models are highly sensitive to changes in body visibility or camera angles. To increase the accuracy of deep learning, the following points are essential.

#### 3.3.1. Creation of New Datasets

There is a need to create newer versions of gait datasets that address real-world challenges. These future datasets should include data captured under diverse weather conditions such as fog, snow, and rain, as these can alter the appearance of the individual. Additionally, the dataset should account for clothing variations, which significantly affect gait patterns. This includes different types of pants, shirts, and both heavy and loose clothing, all recorded in uncontrolled environments. The dataset should also encompass a wide range of multi-view angles, from 0 to 360 degrees, as well as scenarios where individuals are carrying objects, both in hand and on their back. By expanding the training data to include real-world challenging information such as variations in clothing, diverse view angles, occlusion, and carrying objects, the gait models will be better equipped to perform more accurately in complex environments.

#### 3.3.2. Integration of Multimodal Biometrics

To increase the accuracy of the gait identification system, combining multimodal biometrics methods, such as integrating gait identification with facial recognition, can provide a more comprehensive and reliable solution. Gait is a non-intrusive form of identification that is not effective when there are variations in clothing, occlusion, and viewpoint changes. A multimodal system can compensate for the limitations of gait recognition alone. Furthermore, the combination of different biometrics solutions can enhance security by offering more than one form of verification, making it more difficult for the system to be tricked by simple occlusions or changes in viewpoint.

### 3.4. Theoretical Modeling of Covariate Impacts on Gait Recognition

Covariate factors play an important role in gait recognition accuracy. Covariate factors, such as clothing variation, view angle, and walking speeds, can decrease the accuracy of the gait network. It is important to develop a theoretical framework that quantifies their influence. This section presents a conceptual model for understanding the impact of covariates on the gait network.

#### 3.4.1. Influence of View Angles

Changes in the view angle create significant variations in the appearance of the human while walking. From a theoretical point of view, these changes caused by view angles can be modeled using a method called geometric transformations. For example, when a person walks at an angle to the camera, the view of the person’s body or body features changes, along with changes in the input data. The impact of view angles can be described using a model called perspective projection, where 2D frames are the projection of a 3D human body. The accuracy of the gait recognition system is affected by the cosine angle between the walking direction and the camera view.

Mathematically, the change in accuracy can be modeled as:(5)$$Accuracy\left(\theta \right)=A_{O}\times \cos\left(\theta \right)$$
where $A_{o}$ is the accuracy under frontal conditions, and θ is the angle between the camera’s line of sight and the person’s walking direction.

#### 3.4.2. Impact of Clothing Variations

Feature distortion methods are used in machine learning to model clothing variation in gait identification systems. In gait recognition, when a person wears different types of clothing, such as long coats, backpacks, coded shoes, and half-sleeve shirts, the gait patterns of the human change. The variations in clothing are described using the distortion function *D*(*c*), where *c* represents the variation in clothing. The influence of clothing on the gait identification system can be represented by:(6)$$Accuracy\left(c\right)=A_{o}\times e^{-kD\left(c\right)}$$
where *D*(*c*) is the function utilized for quantifying the distortion within gait features caused by variations in clothing, and *k* is a constant that determines how sensitive the model is to these changes.

#### 3.4.3. Influence of Walking Speed

Variation in walking speed affects gait features, such as stride length and step frequency, which can be modeled using kinematic equations from biomechanics. While variations in walking speed occur, the temporal and spatial features in the gait cycle also change. The variation in walking speed can be represented as:(7)$$Accuracy\left(v\right)=A_{o}\times \left(1-\frac{\left|v-v_{o}\right|}{v_{max}}\right)$$
where *Vo* is the normal walking speed, and *Vmax* is the maximum walking speed considered. The accuracy of the model decreases when the walking speed changes from normal walking.

#### 3.4.4. Combining Covariate Factors

The combined effect of different covariates, such as view angles, clothing variations, and walking speeds, can be modeled using a method called the multivariate function.(8)$$Accuracy\left(v,\theta ,c\right)=A_{o}\times e^{-kD\left(c\right)}\times \cos\left(\theta \right)\times \left(1-\frac{\left|v-v_{o}\right|}{v_{max}}\right)$$

The equation combines the effects of multiple covariates, where each factor is weighted according to its influence on the gait recognition accuracy.

## 4. Conclusions and Future Research

Numerous methods have been proposed to address different types of covariate factors and view angles in gait recognition. Notably, successful results have been accomplished through applying numerous deep learning methods and computer vision. Various datasets have been explored, and different evaluation metrics have been employed to assess the test results. The finest gait recognition performance was described in the study [51], where Fine-tuned DenseNet-201 and VGG-16, combined with a Vision Transformer, achieved an average accuracy of above 90%. Similarly, other studies reported excellent results, with average accuracies of 87% [42] and 97% for normal walking [44]. While important advancements have been made in the field, numerous challenges are still essential to be addressed in future research. A dataset is required to have covariates like real-life scenarios, like different varying lighting conditions, low-resolution frames, weather conditions, age variations, hairstyles, and temperature fluctuations. Currently, available datasets only cover a limited range of covariates. For example, the well-known CASIA B gait dataset covers only three covariates: regular walking, walking while holding a bag, and walking dressed in a long coat.

To tackle the challenges of the gait recognition, the dataset contains frames with variable attributes associated with the subject’s pose variant; footwear variations; carrying objects; different walking speeds; different backgrounds; multiple frames per second; Diverse and Global Representation; and Extreme Environmental Conditions like fog, rain, and snow.

### 4.1. Real-World Applications and Deployment Challenges

With rapid advancement in technology, gait recognition systems can be used in various research domains, such as security, law enforcement, healthcare, and retail. In surveillance, gait identification mechanisms are used to identify individuals from a long distance without infringing on their privacy. In smart cities, areas like transportation utilize gait recognition to identify and track individuals for public security purposes. In healthcare, gait recognition is used to detect different types of diseases, such as Parkinson’s and multiple sclerosis, through abnormal gait patterns. In retail, gait recognition systems are used to analyze customer behavior, ensure safety, or manage access to certain sections of large stores. With the advancement in gait recognition, there are still serious challenges and limitations in terms of surveillance conditions, ethical concerns, and scalability that need to be addressed.

#### 4.1.1. Surveillance Conditions

In real time, there are numerous challenges that degrade the accuracy of gait identification systems. A few of them include Covariate challenges, Unconstrained Environment, Spatiotemporal challenges, and Model-based challenges.

***Covariate Challenges:*** Covariates are factors such as clothing, carrying conditions, view angle, and various speeds that decrease gait recognition, regardless of how well your model is trained. However, in real environments, your model faces serious challenges because not all gait datasets cover many covariates, such as a wide range of angles, clothing conditions, and speeds. In the clothing dataset, there are only a few types of shirts and pants, but it does not cover heavy clothing, which changes the walking pattern of the human. While carrying objects, the walking pattern and body structure of the human also affect the accuracy of the gait identification system. The cameras in surveillance are placed at different view angles, which also causes accuracy concerns for the gait identification system.

***Unconstrained Environment:*** In the real world, gait surveillance systems face unconstrained environments such as occlusion, different lighting conditions, and noise, which are still significant challenges. These challenges create difficulties for gait recognition models in accurately extracting gait features from the input frames and affect the performance of the system.

***Spatiotemporal challenges:*** In gait recognition systems, gait energy images are utilized to extract spatial features, but it is very difficult to obtain time-based or temporal features from gait energy images, which makes human recognition less accurate. In temporal feature extraction, recognizing an individual over a long period of time is challenging, as it requires continuous, correct data, and a person’s gait pattern may also change due to aging or injury. The Spatiotemporal challenges emphasize the need for continued research and development to overcome these limitations.

***Model-Based challenges:*** Most model-based methods depend on pose estimation models, which extract different human joint information and pass it to the gait recognition system to extract gait features. However, accurate pose estimation is still a big challenge for frames captured at different view angles and lighting conditions. OpenPose is a famous computer vision tool for pose estimation, but it cannot accurately estimate the pose of gait datasets such as CASIA B and TUM Gait due to variations in view angles. Another challenge in model-based methods is the accurate extraction of human skeleton information when multiple individuals or occlusion are present. When a person’s body is occluded by another person, it becomes impossible to extract skeletal information from the frames.

#### 4.1.2. Ethical and Privacy Concerns

Gait recognition is a biometric identification method that collects people’s gait patterns from a long distance. There is a privacy challenge related to the captured gaits. The problem arises when people’s gait patterns are captured and retained without their familiarity with the technology. These stored gait patterns can be misused in high-risk places, such as airports, hospitals, educational institutions, and public gathering places. This misuse highlights the need for regulatory frameworks that provide guidelines on data collection, usage, and storage, ensuring individuals’ rights are protected.

#### 4.1.3. Scalability Issues

Scaling the gait recognition system on a large scale, such as in smart cities, remains a limitation of the system. The system requires high computational resources and storage capacity to process and store real-time videos. Additionally, the dynamic nature of urban environments, with changing weather conditions and high pedestrian traffic, makes it difficult to maintain consistent accuracy across diverse contexts. With the growth of the population in cities, more efficient real-time algorithms are required that do not compromise on the accuracy and speed of the system. For large deployments in cities, to maintain accuracy and speed, cloud-based and distributed computing techniques are the solution.

#### 4.1.4. Learning Challenges

There are a few learning challenges in deep learning and machine learning techniques for gait recognition. When a model is trained very well on a particular dataset, it creates an overfitting issue. The model performs well on the well-known subjects present in the dataset, but when an unknown subject appears, the model’s performance starts decreasing. Overfitting can be reduced by utilizing data augmentation and regularization techniques. Furthermore, using cross-validation and early stopping during the training process can also minimize overfitting. Deep learning models require a large amount of labeled data during training, which demands significant computational power, time, storage, and the ability to analyze a huge volume of gait patterns. The use of synthetic data is one solution to this problem.

### 4.2. Integration into Existing Surveillance Systems

#### 4.2.1. Technical Integration

The gait identification system can be integrated into existing surveillance systems such as CCTV with the addition of edge computing devices. The edge computing devices are capable of processing real-time video frames and capturing the gait patterns of individuals from a long distance, which enhances the overall functionality and effectiveness of the existing surveillance system. The gait recognition system can be deployed as software within an existing video surveillance system, with videos being passed to the gait system for efficient recognition. This whole system can be implemented using cloud-based or edge computing solutions, where video processing occurs either locally or remotely for real-time gait recognition.

#### 4.2.2. Challenges in Integration

While integrating the gait recognition system into an existing surveillance system, there are a few challenges, as outlined below:

***Data Compatibility:*** Existing surveillance systems use outdated cameras with low resolution and 2D video quality. Gait recognition systems require advanced high-resolution cameras, such as 3D cameras or depth cameras. Model-based methods for gait recognition often require 2D or 3D human pose estimation. Achieving accurate 3D human pose estimation can be costly and challenging with the existing surveillance systems.

***Real-time Processing:*** To process surveillance videos and identify gait patterns from real-time videos, the existing systems are not capable of processing real-time videos without any delay. The addition of real-time processing to an existing system without adding significant latency makes it more complex and expensive. In places like airports and stadiums, the system must be capable of identifying people quickly, and the system cannot bear any delays.

***Scalability:*** When a large-scale surveillance system is deployed in crowded places, it requires more processing power to correctly identify the gait patterns of individuals. The system requires distributed processing across multiple servers or cloud infrastructure, which causes the scalability issue for the system.

#### 4.2.3. Deployment Challenges

***Privacy Concerns:*** Gait recognition systems contain important gait patterns of a large number of individuals, which must be secured from unauthorized users or misuse. The system must also comply with privacy regulations, such as GDPR in Europe and other local privacy laws.

***Environmental Factors:*** Deployment of gait recognition systems in real-time applications faces significant challenges, such as low lighting conditions, diverse weather conditions, clothing variations, view angles, and occlusions. These factors affect the accuracy of the gait recognition system, and systems must be designed to be robust to such challenges.

***Model Training and Adaptation:*** The deployment of gait recognition systems in real-world applications requires model adaptation to local environments. A gait model trained on a specific region and specific people cannot work in another region due to differences in body structure, clothing styles, and environmental conditions. Transfer learning mechanisms can be utilized to improve the performance of the model.

#### 4.2.4. Practical Deployment Scenarios

***Public Spaces and Law Enforcement:*** The gait identification system can be deployed in airports, bus stations, and stadiums for monitoring suspicious behavior or anomaly detection. For example, if an individual is behaving suspiciously, that can be flagged by the system and tracked by multiple cameras while ensuring privacy by only tracking gait patterns.

***Healthcare and Elderly Monitoring:*** In the healthcare sector, gait identification systems are used for monitoring patients’ movements and detecting abnormalities through gait patterns. Changes in gait patterns can indicate different diseases or health issues, such as Parkinson’s and multiple sclerosis disease.

***Retail and Marketing:*** In retail and marketing, gait recognition is used to analyze customer behaviors, such as tracking how customers move through the store, how long they spend in specific sections, and how they interact with products. These gait patterns can be used for targeted marketing and to improve the customer experience.

### 4.3. Actionable Future Research Recommendations

#### 4.3.1. Handling Low-Light Conditions

***Use of Advanced Image Enhancement Techniques:*** To enhance image quality, future research should focus on low-light image restoration or contrast enhancement. Another technique used to enhance the quality of the image is the use of GANs to create high-resolution images from low-light data, which can improve model accuracy under low-light conditions.

***Multi-Modal Data Fusion:*** The combination of infrared, thermal, and traditional RGB images can provide more efficient gait recognition under low-light conditions. Future research must focus on creating datasets that include infrared, thermal, and RGB images, enabling the model to learn better gait features.

***Training with Synthetic Data:*** To handle low-light conditions, the model can be trained on synthetically generated low-light images. With the help of data augmentation techniques like reducing the brightness of the frames or adding noise, the model is capable to learn more efficient features and become more robust for real-world low-light scenarios.

***Improved Sensor Technology:*** Future research should focus on better sensors that can capture detailed information across diverse lighting conditions. These sensors include low-light video sensors or depth sensors. Data from these sensors improve the performance of the model under low-light conditions.

#### 4.3.2. Addressing Occlusions

***Pose Estimation and Body Part Detection:*** Future research should focus on pose estimation techniques that can detect and identify human body parts, even if the subject is partially occluded. Multi-view pose estimation is used to address the occlusion issue by using frames captured from different view angles.

***Use of 3D Modeling and Depth Information:*** When human body parts are partially visible, depth sensing and 3D modeling improve the accuracy of gait recognition models. These techniques collect additional information on human walking patterns and structures, even when some body parts are partially occluded.

***Temporal Feature Fusion:*** Temporal feature extraction from the input frames using techniques like LSTMs or GRUs can improve the accuracy of the gait recognition system, even when human body parts are partially occluded. The system can fill in the missing information of the frames based on the previous frames.

#### 4.3.3. Datasets Needed for Low-Light and Occlusion Handling

***Low-Light Datasets:*** There is a need for publicly available data with varying lighting conditions for gait identification. A model trained on these low-light datasets can perform well in actual low-light conditions. Researchers should create a dataset in which human walking is captured in dark environments. For example, datasets like Night Vision Gait and Thermal Gait could be developed for future studies.

***Occlusion-Focused Datasets:*** The existing gait datasets should include frames with partial occlusion, such as people walking with bags in crowds or being partially hidden. The datasets should also include partially or fully occluded view angles from 0° to 360°, which must be considered in future research. For example, the COCO dataset includes very few frames in which a person’s body parts are partially occluded.

This research can help in security, where person recognition is an essential portion of the latest surveillance systems, as it offers selective access to sites. Human identification is used in many fields, like registration of the daily presence for employees at the workplace; educational institutions; and sensitive locations, such as Air Traffic Control, where there is a specific number of persons who have authorized access. Gait identification is regarded as being among the leading techniques for recognition due to the minimum probability of duplication. Gait recognition also minimizes the margin for human errors and inattention, which are continuously an option in conventional systems. In the future, it will be necessary to design a gait recognition system that reflects real-world scenarios, such as low-light conditions, varying weather, age variations, diverse hairstyles, and temperature fluctuations. Furthermore, advancements in computer vision, which include state-of-the-art object detection approaches including YOLO and spatiotemporal information regarding GANs, should be endowed for improved performance.

## Figures and Tables

**Figure 1 sensors-25-03471-f001:**
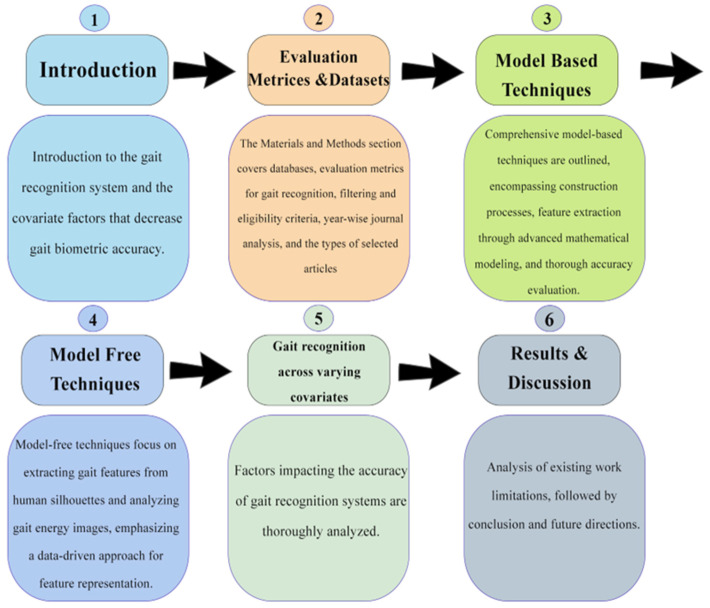
Structural framework for the entire manuscript.

**Figure 2 sensors-25-03471-f002:**
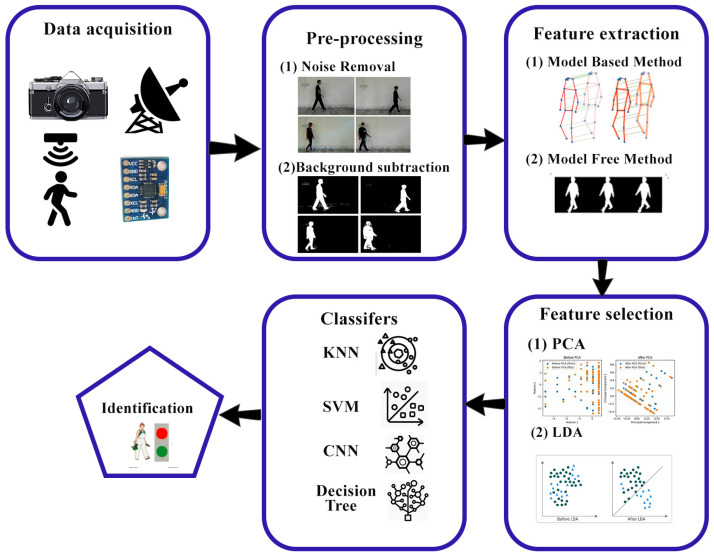
Workflow diagram of Human Gait Analysis System.

**Figure 3 sensors-25-03471-f003:**
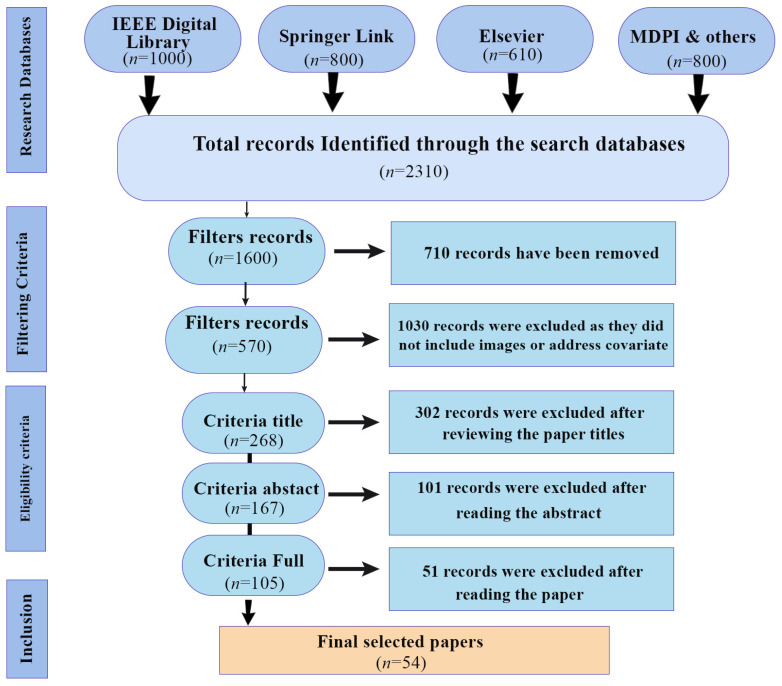
PRISMA flowchart for the selected records.

**Figure 4 sensors-25-03471-f004:**
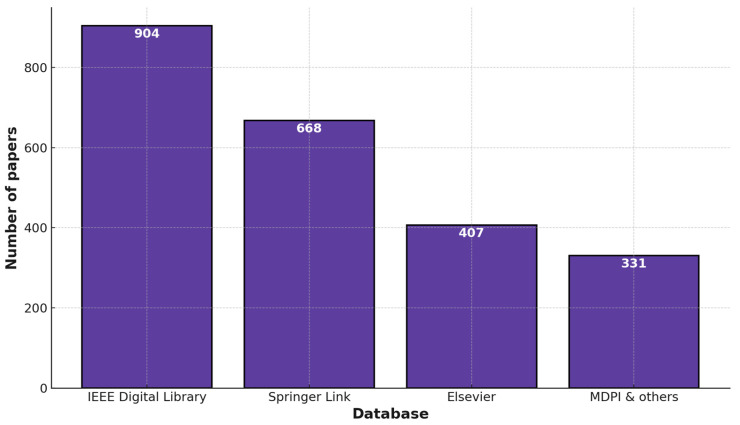
Number of papers on gait recognition from various databases.

**Figure 5 sensors-25-03471-f005:**
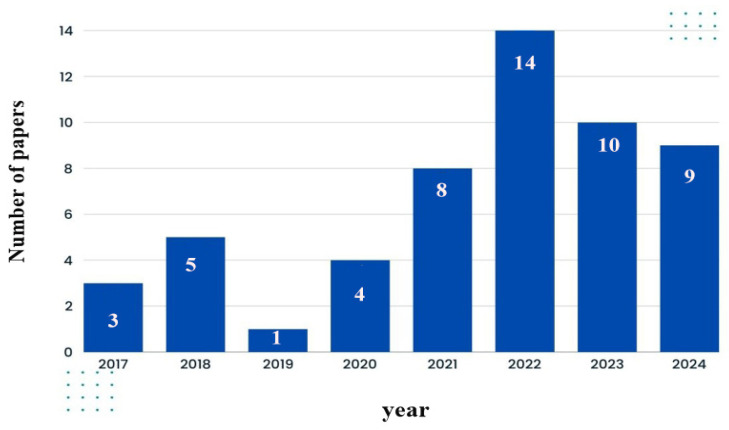
Paper publication frequency by year.

**Figure 6 sensors-25-03471-f006:**
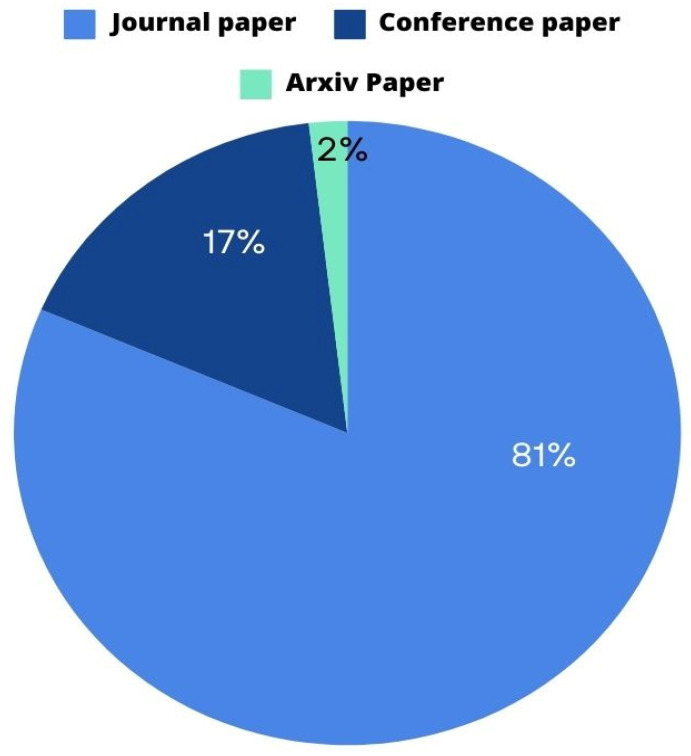
Types of papers selected for our manuscript.

**Figure 7 sensors-25-03471-f007:**
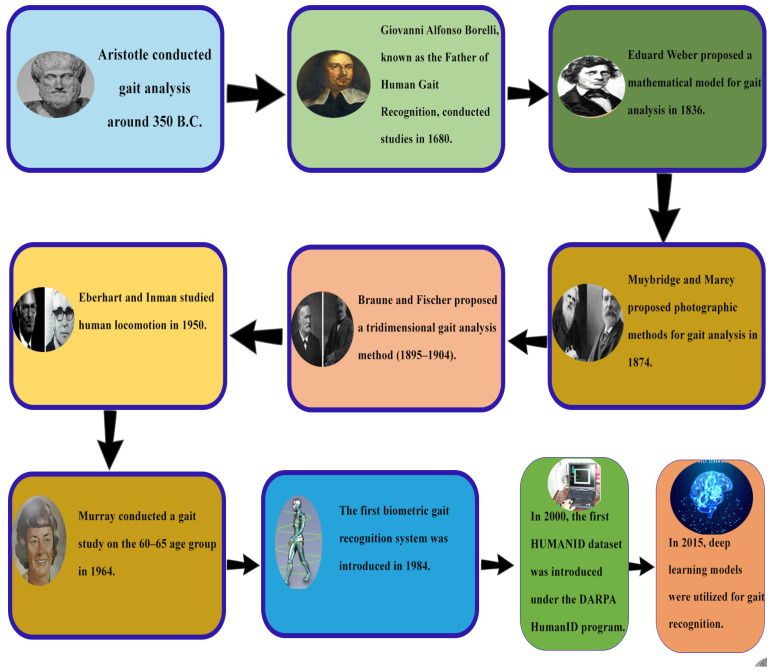
Timeline of gait recognition development to date.

**Figure 8 sensors-25-03471-f008:**
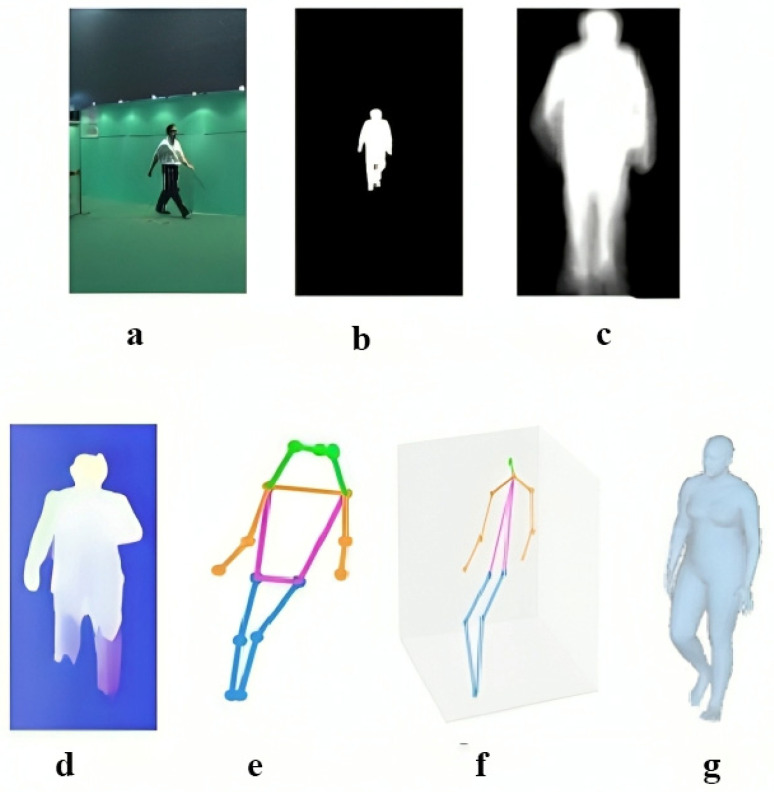
Different input data representations used in gait recognition: (**a**) RGB frame of walking subject; (**b**) Extracted gait silhouette; (**c**) Gait energy image; (**d**) Optical flow indicating motion dynamics; (**e**) 2D skeleton extracted through 2D pose estimation; (**f**) 3D skeleton extracted using 3D pose estimation; (**g**) 3D mesh.

**Figure 9 sensors-25-03471-f009:**
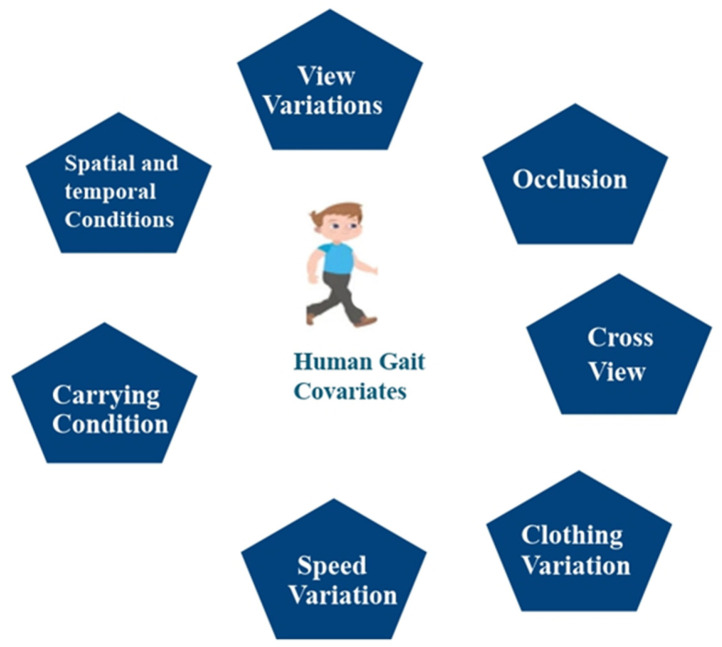
Key covariate conditions that impact gait recognition.

**Figure 10 sensors-25-03471-f010:**
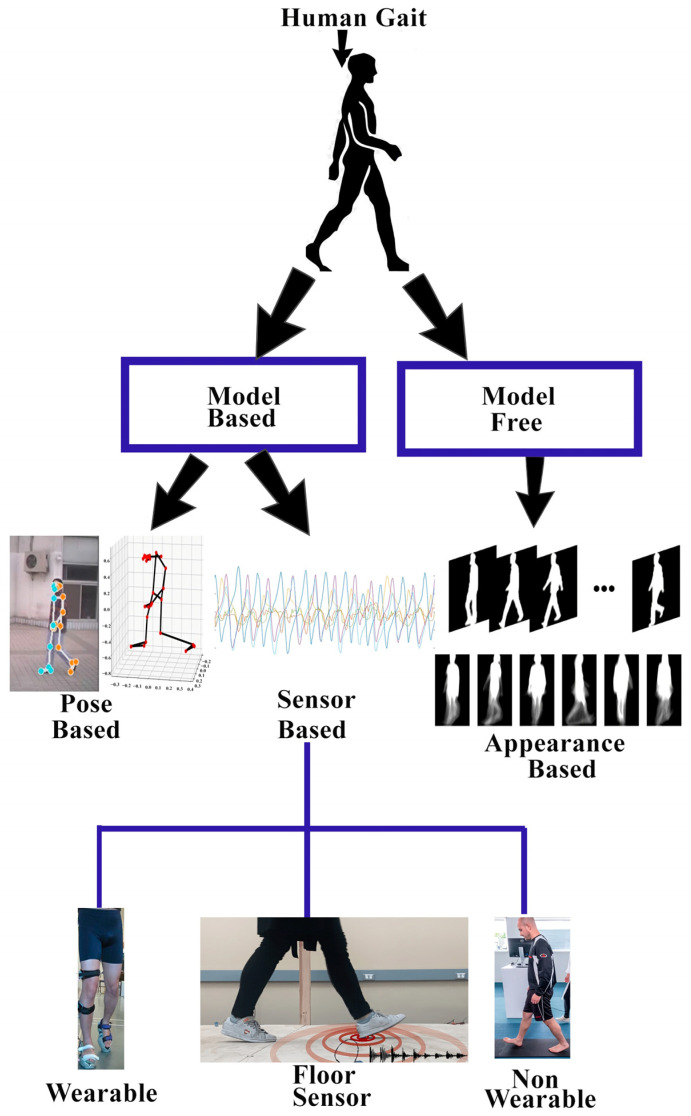
Methods used for gait analysis.

**Figure 11 sensors-25-03471-f011:**
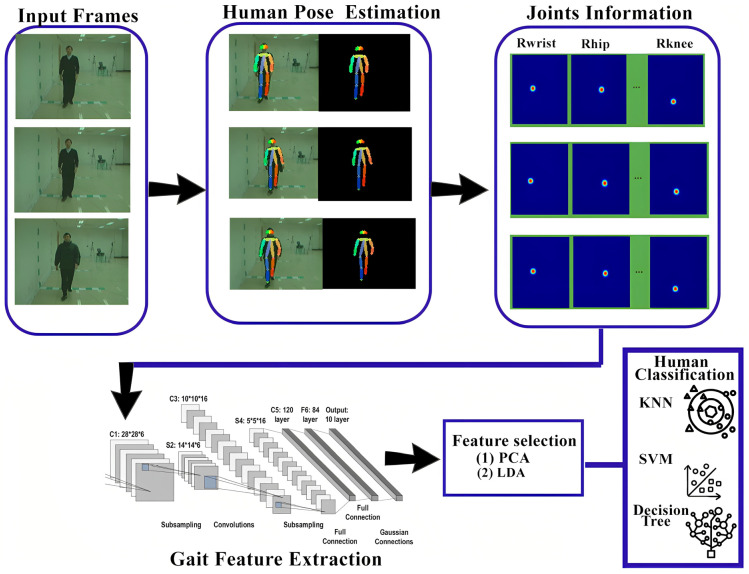
Model-based pipeline.

**Figure 12 sensors-25-03471-f012:**
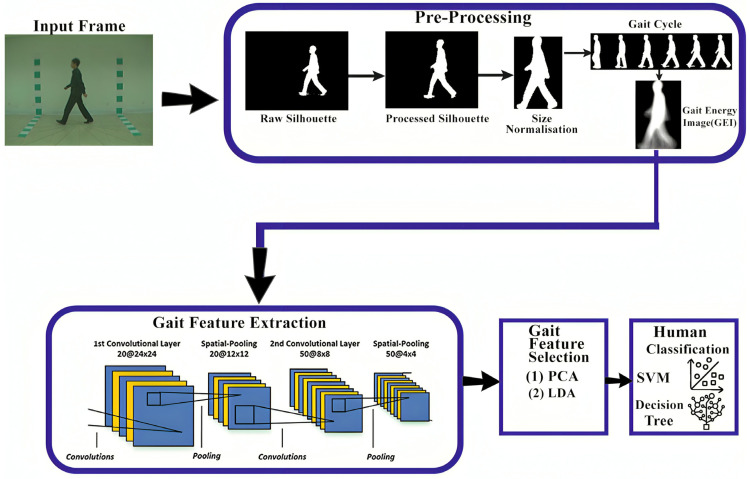
Model-free methodology pipeline.

**Table 1 sensors-25-03471-t001:** Details of the various source datasets for gait recognition.

Datasets	Publication Year	No. of Sequences	No. of Subjects	Images/Videos per Subject (Frames)	Covariate Factors	Image Resolution	Data Type	Frames per Second (FPS)
Tum Gait [23]	2012	3370	305	Multiple	Normal walk, Backpack, long coat, coating shoes	640 × 480	RGB, Depth, Audio	30
CASIA A [11]	2001	19,139	20	Multiple	Normal walk with three view angles: 0°, 45°, and 90°	352 × 240	RGB	25
CASIA B [24]	2005	13,680	124	Not Specified	Normal walk, carrying backpack, wearing jacket or coat, and 11 view angles from 0° to 180°	320 × 240	RGB, Silhouette	25
CASIA C [25]	2005	1530	153	More than 300	Slow walking, Normal walking, regular walking, and fast walking with bag carrying	320 × 240	Infrared, Silhouette	25
CASIA E [26]	2022	778,752	1000	Multiple	Clothing conditions, carrying conditions, walking styles, and 26 different view angles	1920 × 1080	Silhouette	25
OU-ISIR [27]	2009	37,531	4016	1876	Normal walking with four angles: 55°, 65°, 75°, and 85°	64 × 64	Silhouette	30
OU-ISIR treadmill [28]	2007	8728	302	2182	25 different view angles, 32 clothing combinations, different walking speeds from 2 km/h to 10 km/h and gait fluctuations	88 × 128	Silhouette	60
OU-ISIR speed transition [29]	2013	306	34	Not Specified	Different walking speeds from 1 km/h to 5 km/h	88 × 128	Silhouette	60
OU-LP Age [30]	2017	187,584	63,846	Not Specified	Only males and females aged 2 to 90 years old	640 × 480	Silhouette, GEI	30
OU-LP Bag [31]	2018	63,846	2070	Not Specified	Carried objects of seven different types	1280 × 980	Silhouette, GEI	25
OU-MVLP [32]	2016	259,013	10,307	35,840	14 view angles range from 0° to 270° with normal walk	1280 × 980	Silhouette, GEI	25
OU-MVLP Pose [33]	2017	259,013	10,307	35,840	14 view angles with normal walking speed	1280 × 980	2D Skeleton	25
USF [34]	2002	Not Specified	122	Not Specified	32 possible conditions and two viewpoints.	720 × 480	RGB	30
Southampton Gait [13]	2002	1870	122	Not Specified	6 primary covariate conditions include clothing, carrying, walking speed, and view angles	80 × 80	RGB	25
OU-MVLP Mesh [35]	2022	Not Specified	10,307	Not Specified	Normal Walking from 11 different view angles	1280 × 980	3D Human Mesh	25
SOTON [36]	2002	2128	115	Not Specified	Normal Walking speed	Not Specified	Silhouette, RGB	25

**Table 2 sensors-25-03471-t002:** The application of evaluation metrics for each covariate.

Covariate Condition	Rank-1 Accuracy	EER	F1-Score	Precision	Sensitivity	CMC	AUC
View Angles	Decreases with angle variations. It fails to account for side/rear views.	Balance False Acceptance Rate and False Rejection Rate and handle angle variations effectively.	Handles angle variations more effectively compared to Rank-1.	Decreases with angle variations, as it may misidentify.	Sensitivity decreases if the model is not trained on diverse angles.	Effective for evaluating how the system ranks the correct person even with angle differences (top-k).	Effective for evaluating model performance across different view angles.
Clothing Variations	Decreases with clothing change, especially with unseen clothing.	Sensitive to clothing variations. Shows the balance of FAR and FRR.	Helps balance between misidentifications and missed matches.	Decreases when clothing changes, due to misidentification.	Decreases as different clothes may confuse the system.	Helpful for ranking the correct person in top-k positions despite clothing changes.	Effective in assessing model performance despite clothing differences.
Occlusions	Decreases as occlusions block important features, leading to lower accuracy.	Sensitive to occlusions, as it balances false acceptances and rejections.	Decreases if occluded parts of the body lead to missed matches.	Decreases when occlusions lead to false positives.	Decreases if the system misses identifying an occluded person.	Useful for evaluating the correct ranking even with occlusions.	Useful in evaluating the process of how well the model handles occlusions while distinguishing between subjects.
Lighting Conditions	Decreases in low light, as gait features may be harder to extract.	Sensitive to lighting, as the model can struggle to differentiate people in poor light.	Effective when there are lighting variations, as it captures performance across various conditions.	Decreases under poor lighting due to poor image quality.	Decreases when lighting makes features less visible, leading to missed identification.	Useful in showing how often the correct person appears in top-k despite poor lighting.	Effective for analyzing how well the model distinguishes individuals in low-light conditions.
Walking Speed	Decreases if the speed is faster or slower than training data.	Sensitive to speed variations, particularly if the system was not trained at different speeds.	Can balance precision and recall even with varying speeds.	Affected by speed changes, especially at extreme speeds.	Decreases when the walking speed varies significantly from the training data.	Useful for evaluating how well the system ranks the correct person despite speed differences.	Effective for determining how well the model distinguishes gait at various speeds.

**Table 3 sensors-25-03471-t003:** Model-based gait techniques: a comparison of different methods.

Ref	Dataset	Technique	Number of Covariates	Advantages	Disadvantages
Qi et al. [22]	CASIA B	ResNet50 and LSTM	Clothing variation and view angles from 0 degree to 180 degrees.	Lightweight model, and the computational time of the model is very low.	The study depends on the pose estimation technique, which gives errors on low-resolution videos. The accuracy of the model with normal walking is 73%, but when covariates such as clothing variation, carrying conditions, and view angle are introduced, the model’s accuracy decreases to 49%. Furthermore, the model is trained on only one CASIA B dataset and does not handle speed variations and lighting condition issues.
Pan et al. [37]	CASIA A and CASIA B	LUGAN and GCN	The covariates addressed in the research include view angles from the CASIA A dataset (0° to 45°) and different clothing variations, along with hand carry and view angles from the CASIA B dataset.	The LUGAN method reduces the cross-view variance by generating multi-view poses from a single view input, and the integration of the hypergraph convolution module makes the model learn multi-scale features, from the joint level to the body level.	The limitations of the research include computational complexity. The LUGAN technique, along with the pose estimation and hypergraph convolution methods, requires high computational resources. The pose estimation model in the research is a 2D pose method, which does not fully capture body joint information and complex body movements.
Liao et al. [38]	CASIA B	OpenPose and CNN	View angles from the CASIA B dataset range from 0° to 180°, along with clothing variation and hand carry.	The 3D pose estimation method works very well on low-resolution data, and the pose estimation model effectively extracts dynamic motion patterns by using handcrafted and CNN methods.	The accuracy of the methods depends on 3D poses. If the poses are not correctly identified, it will have a negative impact on the model. Although the 3D pose estimation model is efficient, the 2D to 3D pose conversion is computationally expensive. The model only captures information from 14 joints, which is considered limited and does not fully capture the complex dynamic walking patterns.
Liao et al. [39]	MoBo and CASIA B	GaitMap-CNN and GCN	The covariates covered in this research include the MOBO dataset with view angles from 0° to 315°, and the CASIA B dataset with clothing variation, bag carrying, and view angles from 0° to 180°.	Pose estimation maps contain richer information about the human body compared to traditional skeleton-based methods and are much less sensitive to variations in human shape. The fusion of heatmap evaluation features and pose features creates more discriminative gait features.	The recognition accuracy of PoseMAPGait is very low compared to other models, and using the graph convolutional neural network and pose estimation model makes the model resource-intensive for real-time applications. The model’s accuracy decreases on small datasets or datasets with a limited number of subjects.
Luo et al. [40]	CMU MOBO and CASIA B	Hierarchical Temporal Memory (HTM) and Gait semantic folding	The covariates included from the CMU MoBo dataset, which contain different walking speeds such as slow, normal, and fast, as well as carrying objects like ball carrying. Additionally, the CASIA B dataset is used, which contains view angles and clothing variation.	With the integration of Hierarchical Temporal Memory, the model can learn temporal features more effectively and efficiently even if the input frames are noisy or contain changing environments. Memory-based methods such as Hierarchical Temporal Memory can be learned from new input data, which enables the model to perform well on unknown covariates.	The combination of 3D model estimation, semantic folding, and Hierarchical Temporal Memory is computationally expensive, and the model struggles with small datasets that have a limited number of subjects.
Xu et al. [41]	GREW	Graph Convolutional Neural Network (GCN)	View angles and clothing variation.	The model uses a GCN, which creates a relationship between human joints and bones and helps the model accurately capture the spatial and temporal information from the sequences. The model is simplistic compared to other techniques, as it directly embeds 3D skeleton information into graph networks by eliminating the preprocessing steps. The performance of the model on large datasets, such as Grew, is very good.	The model is heavily dependent on 3D pose data, and any error in the pose will negatively affect the entire method. Additionally, the model is computationally intensive due to the use of 3D data, graph convolutions, and embedding networks. The model heavily depended on large datasets for training but struggled on datasets with limited subjects and data.
Choi et al. [42]	UPCVgait and UPCVgaitK2	Centroid-Based Skeleton Alignment and weighted majority voting	The covariates included in this research are view angles and different walking speeds.	The method reduces the error when estimating the position of the joints by aligning the skeleton with the help of the centroid of torso joints, providing better alignment under occlusion. The computational cost is reduced by dividing gait cycles into small patches or phases, and Weighted Majority Voting improves the accuracy rate of the model.	The dual-stage linear matching and quality-adjusted cost matrix introduce complexity and computational overhead, making the method unsuitable for real-time applications and large databases. The method only includes simple gait patterns, and when complex gait patterns are introduced, the method struggles with accuracy. Due to the high frame rate, the method is slow.
Gao et al. [43]	CASIA B	ST-GCN and Canonical Polyadic Decomposition	The covariates included in the research are 11 view angles from 0° to 180°, along with clothing variation and carrying a bag.	The method includes spatial–temporal graph convolutional networks (for spatial and temporal feature extraction, which improves the performance of the model). The addition of Canonical Polyadic Decomposition extracts more robust features and removes redundant features, which makes the model select the most appropriate gait features.	The model extracts spatial and temporal information effectively, but the complexity of the model increases when extracting temporal features from short and incomplete sequences. The CPD removes redundant information from the gait sequences, but the process of decomposing the features can lead to model overfitting. Additionally, the CPD decomposition is computationally expensive.
Wang et al. [44]	CASIA A, CASIA B, and OU-MVLP	Autoencoders and LSTM	The covariates utilized in the research are from the CASIA A dataset, which contains 3 view angles: 0°, 45°, and 90°. Other covariates include view angles from 0° to 180° from the CASIA B dataset, along with clothing variation and carrying conditions. Additional covariates are from the OU-MVLP dataset, which contains view angles from 0° to 270°.	The method uses stacked autoencoders, which convert skeleton joint coordinates from any arbitrary view to a canonical view without the need for prior knowledge of the view angle. This allows the model to work better with different view angles. By using LSTM with the autoencoder, the model is capable of preserving both spatial and temporal information, which is essential for accurate gait identification.	The stacked autoencoder with multiple layers is computationally expensive on large datasets, and the model requires a large number of resources. The normalization process in the preprocessing phase, which adjusts the subject’s skeleton size for camera proximity, creates complexity in the model and affects the accuracy as well.
Teepe et al. [45]	CASIA B	MobileNetV1, Xception, and PCA	The covariates included in the research are from the CASIA B dataset, which includes view angles from 0° to 180°, along with clothing variation and carrying conditions.	The combination of features from two deep learning methods, MobileNetV1 and Xception, enhances the gait recognition accuracy, and the use of PCA increases the computational efficiency of the method. The model achieves high accuracy rates on the CASIA B dataset with an accuracy of 98.77%.	The OAM-SVM classifier may struggle with noisy and undisturbed data, and the use of multiple kernels increases the complexity of the model. As a result, the accuracy of the model drops at large angles, such as 126° and 180°. The use of multiple models, such as MobileNetV1 and Xception, followed by PCA, decreases the computational cost of the system. However, there is limited data, as the model is trained and tested on only 50 individuals from the CASIA B dataset.

**Table 4 sensors-25-03471-t004:** Model-free gait techniques: a comparison of different methods.

Ref	Dataset	Technique	Number of Covariates	Advantages	Disadvantages
Khan et al. [14]	Tum Gait and CASIA B	VGG19 and MobileNet-V2	The covariates included in the research are clothing variation, coated shoes, and backpacks from the TUM Gait dataset, as well as view angles and clothing variation from the CASIA B dataset.	The lightweight deep learning models VGG19 and MobileNet-V2 are utilized, which are very efficient. The Moth Flame Optimization technique is used for the best feature selection, which increases the accuracy of the model and decreases computational time. By using transfer learning techniques to fine-tune VGG19 and MobileNet-V2, the computational costs of the model during training are reduced.	The framework consists of multiple stages, such as pretraining, feature extraction, feature fusion, optimization, and classification, which makes the model more complex and takes longer to implement. Discriminant Canonical Correlation Analysis combines the features of VGG19 and MobileNet-V2, which improves the model accuracy, but it is a time-consuming process when dealing with large datasets.
Asif et al. [51]	CASIA B	CNN and Histogram of Oriented Gradients	The covariates included in the research are from the CASIA B dataset, which includes clothing variation and view angles from 0° to 180°.	The method is effective in multi-view environments, which is very important for real-time applications, and handles the covariate factors efficiently. The Histogram of Oriented Gradients technique is used for feature extraction, which extracts important gait patterns, even under challenging conditions.	This method depends on gait energy images, which are not ideal for low-resolution videos or partially occluded frames. The model performs well on coats, but variations in clothing decrease the model’s accuracy.
Amin et al. [54]	CASIA A, CASIA B, and CASIA C	BiLSTM and YOLOv2	The covariates included in the research are different walking speeds, such as slow walking, normal walking, and fast walking, from the CASIA C dataset, along with 3 view angles (0°, 15°, and 45°) from the CASIA A dataset, and clothing variation, carrying a bag, and view angles from 0° to 180° from the CASIA B dataset.	The Bi-LSTM with the addition of multiple convolutional layers helps the model capture temporal and spatial features, which makes the model more accurate for gait recognition. The addition of YOLOv2 with SqueezeNet architecture makes the localization of the model faster in real-time applications.	The use of Conv-BiLSTM along with YOLO creates complexity while training the model, as it requires high computational resources, especially on large datasets. The model shows high accuracy in some classes, such as clothing wear and slow walk, but the performance decreases on classes like normal walk and carrying a bag, which causes the model to overfit on some specific classes.
Yujie et al. [55]	CMU Mobo	CNN and LSTM	The research includes the covariates from the CMU MoBo dataset, which includes normal walking, backpack walking, walking in a coat, and ball-holding.	The use of CNN and LSTM to extract spatiotemporal features of human gait makes the model robust for variations in movement and conditions. The LSTM parameters are optimized, which improves the accuracy of the model in challenging conditions like clothing, occlusion, and carrying objects. The CNN model ResNet34 is used for spatial feature extraction, which includes residual blocks that help the model learn more robust features.	Deep learning models based on CNN and LSTM require large computational resources for training the model on large datasets, and the model’s performance is dependent on the quality of the input frames. If the input frames are noisy, the performance of the model decreases. Deep learning models overfit small datasets, as they require large datasets for training the model.
Elharrouss et al. [56]	CASIA B, OU-ISIR, and OU-MVLP	Background modeling and CNN	The research covers large covariates: view angles from the OU-MVLP dataset, which range from 0° to 270°, along with clothing variation and view angles from the CASIA B dataset, where the view angle range is from 0° to 180°. The OU-ISIR dataset contains different view angles.	The model uses a background subtraction method for human silhouette extraction, which enhances the accuracy of the gait recognition process. The dual-stage CNN method initially calculates the angle of the input frames, which then becomes the input for the gait recognition model to handle covariates effectively.	The accuracy of the model depends on the correct estimation of the view angle. If the view angle is not estimated correctly, it leads to decreased recognition accuracy. The model requires detailed background modeling to calculate the human silhouette, and the dual CNN network with multiple layers makes the system complex and slower.
Mogan et al. [57]	CASIA B, OU-ISIR, and OU-LP	DenseNet 201, VGG 16, and Vision Transformer	The covariates included in the research are from the CASIA B dataset, which contains 11 view angles, clothing variation, and carrying objects; the OU-ISIR dataset with view angles; and the OU-LP dataset with 4 view angles: 55°, 65°, 75°, and 85°.	The combination of three deep learning models, such as DenseNet 201, VGG 16, and Vision Transformer, allows the model to extract more accurate and robust gait features. The introduction of an ensemble learning module combines every deep learning model output to increase accuracy and reduce variance, resulting in more accurate gait recognition in challenging conditions like in noise and occlusion.	The research heavily depends on pretrained models, which means it relies on their generalization ability. If the model is not fine-tuned properly, the system’s performance degrades. Another disadvantage of the research is that, due to the multiple layers of model integration, fine-tuning, and fusion, the system becomes complex and time-consuming.
Mogan et al. [58]	CASIA B and OU-ISIR	Vision Transformer	The covariates are view angles, clothing variation, and carrying objects.	The research utilized the Gait-ViT technique, which includes a self-attention block that focuses on important gait features while suppressing the impact of occlusion and noise. The Vision Transformer has the ability to encode both local and global information with the help of multi-head self-attention and residual connections.	The Gait-ViT technique requires high computational resources to handle high-resolution images and large datasets. The model requires a high input size of 64 × 64, which increases computational time and memory usage. Additionally, the model has a single convolutional layer, which cannot extract deep gait features.
Castro et al. [60]	CASIA B and Grew	AttenGait	The covariates are view angles from 0° to 180°, along with clothing variation.	The model known as AttenGait utilizes three trainable attention mechanisms: Attention Conv, Spatial Attention HPP, and Temporal Attention HPP, which allow the model to extract the most important regions from the gait data. AttenGait uses optical flow and grayscale images as input, which contain richer data and provide useful features of gait.	The use of three attention mechanisms along with multiple convolutional layers makes the system computationally expensive. The model is heavily dependent on optical flow or rich modalities; the unavailability of such modalities decreases the accuracy of the model. Additionally, optical flow and these modalities contain richer features compared to the human silhouette, which raises privacy concerns in some applications.
Yousef et al. [61]	CASIA B and OU-ISIR	GANs, AlexNet, Inception, VGG16, VGG19, ResNet, and Xception	The covariates are 11 different view angles, along with clothing variation and carrying objects.	The research utilized GANs for data augmentation, which balances the dataset by normalizing the frames and enables the model to train on difficult covariate conditions. Deep learning models such as AlexNet, Inception, VGG, ResNet, and Xception extract the most useful gait features and improve the model accuracy. Important gait features are selected by utilizing methods such as Particle Swarm Optimization and Grey Wolf Optimization.	The method is computationally expensive, as GANs are used for data generation, different deep learning models are used for feature extraction, and advanced feature techniques are also utilized. The GANs training is very important; if there is any noise during training, it will affect the accuracy of the entire system. The model is complex and requires high training time for large datasets.
Nithyakani et al. [62]	CASIA B	Multi-Convolutional Stacked Capsule Network	The research includes covariates such as clothing variation, normal walking, and carrying objects.	The Multi-Convolutional Stacked Capsule Network extracts gait features more efficiently by using multiple convolutional layers, stacked capsule networks, and the inception network, which makes the model extract gait features without reducing dimensionality. The inclusion of CLAHEF during the preprocessing phase removes noise within the input frame and increases the accuracy rate. The model’s computational time is reduced by 51.136% to 59.04% compared to other models.	During preprocessing, CLAHEF is introduced to enhance image quality by reducing noise, but it makes the model more complex. If not performed optimally, it can cause over-enhancement, which distorts the gait features. With the use of multiple deep learning techniques, the model requires high computational resources for training.

**Table 5 sensors-25-03471-t005:** Fusion-based gait techniques: a comparison of different methods.

Ref	Dataset	Technique	Number of Covariates	Advantages	Disadvantages
Yao et al. [66]	CASIA B	SGEI and CNN	The covariates included in the research are clothing variation, normal walking, walking with carrying conditions, and view angles from 0° to 180°.	The paper combines model-based and model-free techniques, and this combination makes the model work effectively on clothing variations and view angles. The Skeleton Gait Energy Image helps in extracting body features that are less susceptible to clothing changes, while the gait energy image captures spatial information, which works better under normal walking conditions.	The research consists of a complex multi-stage architecture, which is computationally expensive for training on large datasets. The accuracy of the model is heavily dependent on the accurate extraction of pose information from the input frames; any inaccuracies in detection cause degradation in the performance of the model.
Lu et al. [65]	CASIA B	Mediapipe and CNN	The covariates included in the research are view angles from 0° to 180° and clothing variation.	The use of fusion techniques increases the model’s performance under various covariate conditions, and the model achieves high accuracy, with 99.6% on the CASIA A dataset and 99.8% on the CASIA B dataset. During the preprocessing phase, silhouette extraction and histogram equalization enhance the quality of input frames, enabling the extraction of better gait features and making the model applicable for real-time applications.	The model depends on high-quality RGB videos for extracting accurate human poses, and the availability of high-quality videos is not feasible in low-light conditions. The use of RGB data as input limits the model’s functionality to only depth or infrared data. The multi-stage model, combining both model-based and model-free methods, increases the computational complexity of the model.
Zhao et al. [68]	Sonton-small, OUMVLP, and CASIA B	GCN and GaitGL	The covariates addressed in the research are view angles from 0° to 180° from CASIA B, 0° to 270° from OU-MVLP, as well as clothing variation and walking speed.	The use of Compact Bilinear Pooling extracts high-level gait features and decreases the computational complexity of the model. The scale normalization method ensures that each gait feature is comparable, which increases the effectiveness and accuracy of the model.	During model training, careful hyperparameter tuning is required. This makes it difficult to apply in situations with limited data or time for model development. In the feature extraction process, GaitGL and GaitGraph are used, but they have limited capability to extract high-level gait features.

**Table 6 sensors-25-03471-t006:** Comparison of research work on gait recognition under different covariates conditions.

Ref	Publication Year	Methodology	Dataset	Results	Limitations
M.A. Khan et al. [14]	2022	Pretrained fMobileNet-V2 and VGG 19	Tum gait and CASIA B	Accuracy in simple cases is above 90%, but on different clothing accuracy decreases to 83%.	Feature extraction is time-consuming and depends on some pretrained models, which can be less effective in some gait recognition problems. The model has not been tested in real-world conditions, and only a few covariates are covered.
M. Asif et al. [51]	2022	HOG for feature extraction and SVM for classification	CASIA B	Accuracy on coat wearing is 87%, and overall accuracy is 83%.	Experiments were performed using only one view angle (90 degrees); the other angles were not discussed.
M. Bukhari et al. [52]	2020	LDA for dimensionality reduction, along with CNN and HOG	OU_ISIR	The accuracy under normal walking conditions is 97%, but it decreases on carrying bags or wearing a coat.	The method removes the covariate-affected parts of the body, which also leads to the removal of useful information from the image. Additionally, the research does not focus on view angles.
J. Amin et al. [12]	2021	ResNet-18 and BiLSTM to extract features, along with YOLO	CASIA A, B, and C	The accuracy on the normal walking class and walking with a bag is low.	The model depends on selected features, and some useful features are ignored, leading to a decrease in accuracy at low-quality resolution.
R. Liao et al. [47]	2022	Pose estimation using CNN, and gait features extracted with STGCN	MOBO and CASIA B	The accuracy is low for carrying a bag (58% and 39%) and wearing a coat (41%), as well as for carrying a ball.	During pose estimation, only a few joints are covered, and these joints are not normalized. Additionally, the method does not focus on view angles.
R. Liao et al. [54]	2020	3D poses and spatiotemporal features extracted through CNN	CASIA B and CASIA E	The accuracy is low for carrying a bag and wearing a coat, with rates of 42% and 31%, respectively.	The method can detect the pose of a human when facing the camera but cannot estimate the correct pose when the human is not camera-facing or when parts of the body are occluded.
O. Elharrouss et al. [56]	2021	Angle estimation and gait verification performed using CNN with a SoftMax classifier	OU-MVLP, OU-ISIR, and CASIA B	The average accuracy across different view angles, from 0 degrees to 270 degrees, is approximately 85%.	The research does not contain detailed covariate conditions or, rather, includes only the view angles; the research does not include control environmental factors such as temperature and lighting during the experiment.
J.N. Mogan et al. [57]	2023	Fine-tuned DenseNet-201 and VGG-16, combined with Vision Transformer for gait recognition	OU-ISIR D, OU-ISIR Large Population, and CASIA B	The overall accuracy of the model is excellent on the CASIA dataset, which is about more than 90%.	The research does not provide detailed knowledge of the covariates or view angles on which it works, and the research methodology does not include any techniques to handle low-light or occluded frames.
Y.J. Qi et al. [80]	2022	Model-based approach LSTM and residual network 50 to extract spatial and temporal features from images	CASIA B and OU-ISIR	The overall accuracy of the model is low compared to other models, with 73% view angles from 36° to 144°.	The research contains only one covariate, normal walking, and covers view angles from 36° to 144°. Many view angles are missing, and wearing conditions and walking speed are not included.
M.M. Hasan et al. [46]	2021	Stacked auto encoders for feature extraction, along with a SoftMax classifier	CASIA A and CASIA B	Focused on CASIA B view angles and three covariates’ conditions, achieving an average accuracy of almost 50%.	The research did not focus on walking speeds, viewing angles greater than 180 degrees, and different lighting and weather conditions.
A. Mehmood et al. [48]	2024	Feature extraction using stacked encoder with KNN, BG trees, and SVM classifiers	CASIA B	Only six view angles are in the study, from 0 to 90 degrees. The result on these view angles is 98 percent accuracy.	Only a few angles are utilized. Different walking speeds and clothing variations are not discussed.
F.M. Castro et al. [60]	2024	AttenGait technique to learn deep gait features	GREW and CASIA B	92 percent average accuracy on the CASIA B dataset and 89 percent on the GREW dataset.	The model only extracts local spatial features; when there is a change in position or shape, the model is unable to handle it, and there is also a high computational cost.
R.N. Yousef et al. [61]	2024	GANs with pretrained convolutional neural networks	CASIA B and OU-ISIR	Simple method with 99% accuracy; no covariates and view angles are discussed.	No covariates are discussed, such as view angle, speed variations, and clothing variations, and there is also training instability in the method.
P. Nithyakani et al. [62]	2024	Deep stacked multi-convolutional capsule network	CASIA B	Accuracy on normal walking is good, but it decreases when walking with a coat or walking with a bag.	Only 3 wearing conditions are discussed; view angles, walking speed, and low lighting conditions are not discussed, and the computational time is very high.
M.A. Khan et al. [63]	2023	CNNs and Bayesian Network	CASIA B and CASIA C	The average accuracy on different view angles of the CASIA B dataset is 86 percent, and on the CASIA C dataset, it is 91 percent.	Due to the fusion of the features, the computational time is very high, and the view angles are only discussed from 0 to 180 degrees.
C. Meng et al. [64]	2023	The fusion strategy, integrated with the reconstruction of the human body	CASIA B and Outdoor gait	The accuracy of the outdoor gait dataset is 80%, while the CASIA B dataset also achieves an accuracy of 80%.	The model faces the issue of scalability and does not discuss different views and walking speeds.
X. Huang et al. [8]	2022	The STAR technique for human gait feature extraction	CASIA B and OU-MVLP	The accuracy of the model on CASIA B is 87%, and 89% on the OU-MVLP dataset.	The accuracy of the model is quite low compared to other models, and no discussion is provided on clothing variation or walking speed.
L. Yao et al. [66]	2021	Two branches of Multi-stage CNN	CASIA B	The accuracy is very low across different view angles, ranging from 0 degrees to 180 degrees.	The model’s accuracy is very low on the CASIA B dataset, and other covariates are not discussed in the research. Only a few view angles are considered.
M. Shopon et al. [49]	2021	Graph Convolutional Neural Network with OpenPose model	CASIA B	The average accuracy of the model on normal walking, walking with a bag, and walking with a coat is 91%.	The research does not perform experiments on low-quality data, and different walking speeds and view angles are not discussed.
M.H. Khan et al. [36]	2018	Codebook method, along with Fisher vector encoding and SVM	CASIA A and Tum gait	The ACR is 97% on the TUM Gait dataset and 100% on the CASIA A dataset.	In this research, handcrafted features are utilized, which are not important features. However, only a few covariates are addressed in the TUM Gait dataset, and factors such as view angles, walking speeds, and low-resolution data are not discussed.

**Table 7 sensors-25-03471-t007:** Comparison of traditional methods with advanced deep learning techniques.

Method Type	Technique	Dataset	Covariates Handled	Accuracy (%)	Remarks
Traditional	PCA + LDA + k-NN	CASIA Gait	Minimal	90%	High under clean conditions
Traditional	KSOM + View Transformation	CASIA B	Clothing, View Angle	57%	Low performance under covariates
Traditional	GEI + 2D Pose + LSTM	CASIA B	Clothing, View	68%	Moderate improvement using pose
Traditional	3D Pose + ResNet + LSTM	CASIA B	View, Pose	38%	Struggles with intra-class variation
Traditional	Static + Dynamic Feature Fusion + k-NN	Kinect Skeleton	View Angles	60%	Low generalization
Traditional	CNN (3 Conv Layers) + SVM	OU-ISIR Treadmill B	Clothing	94% (Normal), 60% (Heavy Clothing)	Sensitive to clothing changes
Deep Learning	MobileNet-V2 + VGG19 + DCA	CASIA B, TUM Gait	Clothing, View Angle	90%	Transfer learning improves robustness
Deep Learning	DenseNet + Vision Transformer	OU-ISIR, CASIA B	View Angle	92%	High performance across datasets
Deep Learning	ResNet + BiLSTM	CASIA A/B/C	Clothing	89%	Robust spatiotemporal learning
Deep Learning	EfficientNet + Bayesian Fusion	CASIA B/C	View Angle	86–91%	High accuracy: computational cost noted

**Table 8 sensors-25-03471-t008:** The effect of covariate factors on the accuracy of different models.

Study	Covariate Factor	Normal Walking Accuracy (%)	Changed Condition Accuracy (%)
Ali et al. [81]	Clothing variation, Walking speed	90.33	87 (heavy coat), 93 (fast walking), 94 (slow walking)
Liao et al. [38]	Walking with backpack, Hand carry	63.0	42 (backpack), 31 (hand carry)
Li et al. [82]	Walking with bag, Hand carry	98.0	93 (walking with bag), 80 (walking with hand carry)
Marín-Jiménez et al. [83]	Backpack, Coated shoes	82.0	68 (backpack), 76 (coated shoes)
Su et al. [84]	View angles	99.0	74 (CASIA B 0–180°), 57 (OU-MVLP 0–255°)
Bukhari et al. [52]	Clothing variation, Walking speed	97.0	95 (coat), 91 (with bag), 83 (fast walking), 85 (slow walking)
Fendri et al. [85]	Carrying shoulder bag, Backpack, Handbag, Wearing coat	89.0	79 (shoulder bag), 73 (backpack), 65 (handbag), 73 (coat)
Junaid et al. [53]	Wearing Coat, Walking speed variations	98	81 (CASIA B walking with bag), 88 (CASIA C)
Ali et al. [83]	Hand carry and long coat	95	83 (walking with bag), 58 (walking with long coat)
Wang et al. [86]	View angles, Clothing variations, and Carrying conditions	89 on Grew and 98 on CASIA B	69 (Gait 3D view angles and loathing variations), 89 (CASIA B clothing variations)

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
