# Peer review of "Person Recognition via Gait: A Review of Covariate Impact and Challenges"

_sensors, 2025, doi:10.3390/s25113471_

Round 1
Reviewer 1 Report
Comments and Suggestions for Authors
This paper (Manuscript ID sensors-3489244) presented a review of person recognition via human gait. The review is written in a relatively superficial manner, without in-depth categorization and analysis of the surveyed algorithms. The images used are directly taken from the referenced papers, and the algorithms from the literature have not been reproduced or executed. The authors are highly recommended to address the following comments:
1) The descriptions of the various model methods in the text are relatively simplistic.
2) The introduction to the paper screening process in Section 2 is not strongly connected to the main topic of this paper.
3) The data provided in Section 2.5.1 does not match the data shown in Figure 5.
4) Figure 12 needs to include content that reflects the relevant covariates.
5) A comparison of various methods needs to be provided, and an analysis of the situations in which each method is applicable should be conducted.

Author Response
Thank you very much for taking the time to review this manuscript. Please find detailed responses below and the corresponding revisions/corrections highlighted/in track changes in the re-submitted files.
Comment 1: The descriptions of the various model methods in the text are relatively simplistic
Response 1: We have made revisions to the manuscript based on the reviewer’s suggestions and have included detailed descriptions of all the models used for gait identification. The updated content can be found as follows: on page 19 under the heading '2.8.1 Model-Based Techniques,' on page 26 under '2.8.2 Model-Free Techniques,' and on page 35 under '2.8.3 Fusion Model.' These sections provide a comprehensive overview of the models incorporated in our study.
Comment 2: [The introduction to the paper screening process in Section 2 is not strongly connected to the main topic of this paper]
Response 2: We have revised the screening process and identified some discrepancies in the numbers presented within Figures 3, 4, and 5. We have corrected the values in these figures to ensure accuracy.
Comment 3: [The data provided in Section 2.5.1 does not match the data shown in Figure 5]
Response 3: In the previous version of the paper, there were mistakes in the values presented. We have corrected all the values in Section 2.5.1 to align with the accurate data shown in Figures 5.
Comments : [Figure 12 needs to include content that reflects the relevant covariates.]
Response 4: The details of Figure 12 have been added on page 18 of the paper, where each covariate is discussed in detail.
Comment 5: [A comparison of various methods needs to be provided, and an analysis of the situations in which each method is applicable should be conducted. ]
Response 5: A detailed comparison of model-based, model-free, and fusion-based techniques is discussed in the paper from pages 19 to 37. We also compare which methods are applicable under different conditions and contrast them with traditional machine learning methods.
Reviewer 2 Report
Comments and Suggestions for Authors
Here are my comments:
- There isn't a clear issue definition or explanation of how this review helps the area in the abstract. Indicate clearly the important contributions, the research need (e.g., limitations of previous reviews), and a summary of the results.
- Although background information is given in the opening, the review is not appropriately framed in light of recent developments. Rewrite the introduction in the manner shown below: Clearly describe the significance of gait recognition. Describe the difficulties with covariates and Give a brief overview of previous evaluations and explain how this study is different.
- The manuscript describes the paper selection process (PRISMA flowchart) but does not justify why certain studies were included or excluded. Recommendation: Provide specific reasons for: Why certain covariate factors were prioritized; Why studies using sensor-based gait recognition were excluded and Whether studies before 2017 were omitted due to outdated techniques.
- Although the paper enumerates many gait identification methods, it does not thoroughly examine their advantages and disadvantages. Add a table that compares and summarizes: the application of several methods; Which models exhibit optimal performance under particular factors and the compromises between model-based and model-free methods.
- While descriptive, the consideration of covariate variables (such as attire, walking pace, and occlusion) is shallow. Extend the analysis by: Quantifying the performance declines caused by each covariate; Talking about how various datasets manage covariates and contrasting the models that are most resistant to particular factors.
- The manuscript does not discuss practical challenges in real-world gait recognition systems. Address real-world limitations, such as: Surveillance conditions (low-light, occlusions); Ethical concerns (privacy in public spaces) and Scalability issues (large-scale deployment in smart cities).
- Although previous study findings are included in the publication, contradictions in the findings are not thoroughly evaluated. Discuss the following topics: how experimental circumstances impact performance comparisons; why some studies show inconsistent accuracy rates; and what future benchmarks ought to be applied for equitable comparisons.
- The section on future research is very broad and fails to identify important constraints of the models that are already in use. Talk about: Why deep learning techniques still have trouble with occlusion and multi-view scenarios; how new datasets should be created (e.g., varied weather conditions, outfit changes); and Possible combination of multimodal biometrics (e.g., facial recognition and gait).
The English could be improved to more clearly express the research.
Author Response
Thank you very much for taking the time to review this manuscript. Please find detailed responses below and the corresponding revisions/corrections highlighted/in track changes in the re-submitted files.
Comments 1: [There isn't a clear issue definition or explanation of how this review helps the area in the abstract. Indicate clearly the important contributions, the research need (e.g., limitations of previous reviews), and a summary of the results.]
Response 1: The abstract of the paper has been updated in accordance with the suggestions provided
Comments 2: [Although background information is given in the opening, the review is not appropriately framed in light of recent developments. Rewrite the introduction in the manner shown below: Clearly describe the significance of gait recognition. Describe the difficulties with covariates and Give a brief overview of previous evaluations and explain how this study is different..]
Response 2: The introduction of the paper has been revised in line with the suggestions provided.
Comments 3: [The manuscript describes the paper selection process (PRISMA flowchart) but does not justify why certain studies were included or excluded. Recommendation: Provide specific reasons for: Why certain covariate factors were prioritized; Why studies using sensor-based gait recognition were excluded and Whether studies before 2017 were omitted due to outdated techniques.]
Response 3: The explanation is provided in Section 2.4, 'Inclusion Criteria,' on page 7.
Comments 4: [Although the paper enumerates many gait identification methods, it does not thoroughly examine their advantages and disadvantages. Add a table that compares and summarizes: the application of several methods; Which models exhibit optimal performance under particular factors and the compromises between model-based and model-free methods]
Response 4: A comparison table of model-based, model-free, and fusion-based methods has been created. The details of the model-based methods are provided in Section 2.8.1 'Model-Based Techniques' on pages 19 to 26, the model-free techniques are detailed in Section 2.8.2 'Model-Free Techniques' on pages 26 to 35, and the fusion-based techniques are covered in Section 2.8.3 'Fusion Model' on pages 35 to 38. In the results section, the details and results of each paper are analyzed in Table 6.
Comments 5: [While descriptive, the consideration of covariate variables (such as attire, walking pace, and occlusion) is shallow. Extend the analysis by: Quantifying the performance declines caused by each covariate; Talking about how various datasets manage covariates and contrasting the models that are most resistant to particular factors.]
Response 5: On page 42, Section 3.3 'Quantitative Covariate Impact and Model Robustness Comparison' is discussed in detail.
Comments 6: The manuscript does not discuss practical challenges in real-world gait recognition systems. Address real-world limitations, such as: Surveillance conditions (low-light, occlusions); Ethical concerns (privacy in public spaces) and Scalability issues (large-scale deployment in smart cities).]
Response 6: In the Conclusion and Future Work section, the details of Section 4.1 'Real-World Applications and Deployment Challenges' are discussed.
Comments 7: Although previous study findings are included in the publication, contradictions in the findings are not thoroughly evaluated. Discuss the following topics: how experimental circumstances impact performance comparisons; why some studies show inconsistent accuracy rates; and what future benchmarks ought to be applied for equitable comparisons.
Response 7: A detailed discussion on Section 3.2 'Inconsistencies in Prior Findings and the Need for Standardized Benchmarks' is provided on page 4.
Comments 8: The section on future research is very broad and fails to identify important constraints of the models that are already in use. Talk about: Why deep learning techniques still have trouble with occlusion and multi-view scenarios; how new datasets should be created (e.g., varied weather conditions, outfit changes); and Possible combination of multimodal biometrics (e.g., facial recognition and gait).
Response 8: Below Table 8 on page 44, a detailed discussion is provided on why deep learning models still struggle with covariate factors, followed by Section 3.3.1 'Creation of New Datasets' and Section 3.3.2 'Integration of Multimodal Biometrics.
Reviewer 3 Report
Comments and Suggestions for Authors
The paper provides a comprehensive review of the current state of gait recognition, focusing on the impact of covariate factors such as view angles, clothing variations, walking speeds, and occlusion. The authors analyze various methods used in gait recognition, including model-based, model-free, and hybrid approaches, and discuss their strengths and limitations. However, the paper still has the following problems:
The paper provides a broad overview of various gait recognition methods but lacks detailed implementation details for the models discussed. For instance, the specific architectures of the neural networks used in deep learning-based methods are not elaborated. Providing pseudocode or references to open-source implementations would enhance the reproducibility of the methods discussed.
Limited Evaluation and Ablation Studies: The review mentions several studies with high accuracy on specific datasets, but there is limited discussion on ablation studies that could provide insights into the contributions of individual components of the models. For example, how does the performance change when different covariate factors are considered or excluded? Including such analyses would strengthen the understanding of the methods' robustness.
The paper discusses the impact of covariate factors on gait recognition but does not provide a theoretical framework for quantifying these impacts. For instance, how can the influence of view angles or clothing variations be theoretically modeled? Adding a section on theoretical analysis could provide a deeper understanding of the problem.
The review mentions various methods and their performance but lacks a comprehensive comparison with well-known baselines in the field. For example, how do the reviewed methods compare to traditional computer vision techniques or simple machine learning models? Including a comparative analysis would help highlight the advancements brought by recent deep learning methods.
The paper occasionally uses technical jargon without sufficient explanation, which may make it difficult for readers unfamiliar with the field to understand. For example, terms like "spatio-temporal features" and "graph convolutional networks" are mentioned without clear definitions. Providing brief explanations or glossaries for such terms would improve clarity.
The paper focuses on the technical aspects of gait recognition but lacks detailed discussion on real-world applications and deployment challenges. For instance, how can gait recognition systems be integrated into existing surveillance systems? Discussing practical deployment scenarios would make the review more applicable to industry practitioners.
The paper mentions several evaluation metrics used in gait recognition but does not provide a critical analysis of their suitability for different scenarios. For example, how do metrics like Rank-1 accuracy and Equal Error Rate (EER) compare in terms of their ability to reflect real-world performance? A detailed discussion on the choice of metrics would be beneficial.
The paper outlines future research directions but lacks specific recommendations for addressing the identified challenges. For example, what specific techniques or datasets are needed to handle low-light conditions or occlusions? Providing actionable suggestions would guide future research efforts more effectively.
The review mentions several datasets used in gait recognition but does not critically analyze their limitations. For instance, how representative are these datasets of real-world scenarios? Discussing the limitations of existing datasets and suggesting improvements would help in developing more robust gait recognition systems.
The papers in the introduction of the paper are old and insufficient, and the background description needs to cite more papers. The following paper needs to be cited: "From Sample Poverty to Rich Feature Learning: A New Metric Learning Method for Few-Shot Classification"
Author Response
Thank you very much for taking the time to review this manuscript. Please find detailed responses below and the corresponding revisions/corrections highlighted/in track changes in the re-submitted files.
Comments 1: [The paper provides a broad overview of various gait recognition methods but lacks detailed implementation details for the models discussed. For instance, the specific architectures of the neural networks used in deep learning-based methods are not elaborated. Providing pseudocode or references to open-source implementations would enhance the reproducibility of the methods discussed.]
Response 1: The detailed architecture of the techniques used in model-based, model-free, and fusion-based methods is discussed in the paper from pages 19 to 38, under Section 2.8.1 'Model-Based Techniques,' Section 2.8.2 'Model-Free Techniques,' and Section 2.8.3 'Fusion Model.
Comments 2: [Limited Evaluation and Ablation Studies: The review mentions several studies with high accuracy on specific datasets, but there is limited discussion on ablation studies that could provide insights into the contributions of individual components of the models. For example, how does the performance change when different covariate factors are considered or excluded? Including such analyses would strengthen the understanding of the methods' robustness.s.]
Response 2: The impact of covariate factors is discussed in detail in Section 3.3 'Quantitative Covariate Impact and Model Robustness Comparison' on page 4.
Comments 3: [The paper discusses the impact of covariate factors on gait recognition but does not provide a theoretical framework for quantifying these impacts. For instance, how can the influence of view angles or clothing variations be theoretically modeled? Adding a section on theoretical analysis could provide a deeper understanding of the problem]
Response 3: The detailed discussion on Section 3.4 'Theoretical Modeling of Covariate Impacts on Gait Recognition' is provided on page 44.
Comments 4: [The review mentions various methods and their performance but lacks a comprehensive comparison with well-known baselines in the field. For example, how do the reviewed methods compare to traditional computer vision techniques or simple machine learning models? Including a comparative analysis would help highlight the advancements brought by recent deep learning methods.]
Response 4: The detailed discussion on Section 3.1 'Comparison with Traditional Baselines' is provided on page 40, where traditional machine learning methods are compared with advanced deep learning methods.
Comments 5: [The paper occasionally uses technical jargon without sufficient explanation, which may make it difficult for readers unfamiliar with the field to understand. For example, terms like "spatio-temporal features" and "graph convolutional networks" are mentioned without clear definitions. Providing brief explanations or glossaries for such terms would improve clarity..]
Response 5: The terms are defined where they are used in the paper.
Comments 6: [The paper focuses on the technical aspects of gait recognition but lacks detailed discussion on real-world applications and deployment challenges. For instance, how can gait recognition systems be integrated into existing surveillance systems? Discussing practical deployment scenarios would make the review more applicable to industry practitioners. ]
Response 6: In the Conclusion and Future Work section, a detailed discussion is provided on Section 4.1 'Real-World Applications and Deployment Challenges,' Section 4.2.2 'Challenges in Integration,' and Section 4.2.4 'Practical Deployment Scenario.
Comments 7: [The paper mentions several evaluation metrics used in gait recognition but does not provide a critical analysis of their suitability for different scenarios. For example, how do metrics like Rank-1 accuracy and Equal Error Rate (EER) compare in terms of their ability to reflect real-world performance? A detailed discussion on the choice of metrics would be beneficial.]
Response 7: The evaluation metrics are discussed in detail in Section 2.7 'Evaluation Metrics,' and the table provides a discussion on which evaluation metrics are suitable for each covariate.
Comments 8: [The paper outlines future research directions but lacks specific recommendations for addressing the identified challenges. For example, what specific techniques or datasets are needed to handle low-light conditions or occlusions? Providing actionable suggestions would guide future research efforts more effectively]
Response 8: On page 49, Section 4.3 'Actionable Future Research Recommendations' provides a detailed discussion on future directions for handling covariate conditions.
Comments 9: [The review mentions several datasets used in gait recognition but does not critically analyze their limitations. For instance, how representative are these datasets of real-world scenarios? Discussing the limitations of existing datasets and suggesting improvements would help in developing more robust gait recognition systems.]
Response 9: A detailed discussion on the datasets, along with the limitations of each dataset, is provided in Section 2.6 'Source Database' and Section 2.6.1 'Limitations of Existing Datasets and Recommendations.
Comments 10: [The papers in the introduction of the paper are old and insufficient, and the background description needs to cite more papers. The following paper needs to be cited: "From Sample Poverty to Rich Feature Learning: A New Metric Learning Method for Few-Shot Classification.]
Response 10: The introduction of the paper has been updated according to the respective reviewers' suggestions.
Round 2
Reviewer 3 Report
Comments and Suggestions for Authors
I have no further comments.